# Biporous silica nanostructure-induced nanovortex in microfluidics for nucleic acid enrichment, isolation, and PCR-free detection

Eunyoung Jeon [1,2,3,9], Bonhan Koo [4,9], Suyeon Kim [1,2,3,9], Jieun Kim[5], Yeonuk Yu[5], Hyowon Jang[6], Minju Lee[4], Sung-Han Kim[7], Taejoon Kang [6], Sang Kyung Kim[8], Rhokyun Kwak[5] ✉, Yong Shin [4] ✉ & Joonseok Lee [1,2,3] ✉

Efficient pathogen enrichment and nucleic acid isolation are critical for accurate and sensitive diagnosis of infectious diseases, especially those with low pathogen levels. Our study introduces a biporous silica nanofilms-embedded sample preparation chip for pathogen and nucleic acid enrichment/isolation. This chip features unique biporous nanostructures comprising large and small pore layers. Computational simulations confirm that these nanostructures enhance the surface area and promote the formation of nanovortex, resulting in improved capture efficiency. Notably, the chip demonstrates a 100-fold lower limit of detection compared to conventional methods used for nucleic acid detection. Clinical validations using patient samples corroborate the superior sensitivity of the chip when combined with the luminescence resonance energy transfer assay. The enhanced sample preparation efficiency of the chip, along with the facile and straightforward synthesis of the biporous nanostructures, offers a promising solution for polymer chain reaction-free detection of nucleic acids.

Nucleic acids (NAs) serve as diagnostic biomarkers in accurate and sensitive diagnosis of various diseases including cancer[1–3], genetic disorder[4], and infectious diseases[5,6]. Particularly in infectious diseases, molecular diagnostics, such as polymerase chain reaction (PCR)-based detection, are ideal for early diagnosis due to the swift and substantial accumulation of the viral genome compared to the slower antibody response following symptom onset[7,8]. However, during the asymptomatic or pre-symptomatic period[7,9], the low concentration of the pathogen poses challenges to detect without sample preparation, such as pathogen and nucleic acid enrichment/isolation.

Commercial NA extraction methods primarily depend on surface binding, in which NAs adhere to solid surfaces of columns, beads, or microfluidic channels[10]. However, spin-column or bead-based platforms are laborious and offer limited sample volume[11,12]. Conversely,

[1]Department of Chemistry, Hanyang University, Seoul 04763, Republic of Korea. [2]Research Institute for Natural Science, Hanyang University, Seoul 04763, Republic of Korea. [3]Research Institute for Convergence of Basic Sciences, Hanyang University, Seoul 04763, Republic of Korea. [4]Department of Biotechnology, College of Life Science and Biotechnology, Yonsei University, Seoul 03722, Republic of Korea. [5]Department of Mechanical Convergence Engineering, Hanyang University, Seoul 04763, Republic of Korea. [6]Bionanotechnology Research Center, Korea Research Institute of Bioscience and Biotechnology (KRIBB), Daejeon 34141, Republic of Korea. [7]Department of Infectious Diseases, Asan Medical Center, University of Ulsan College of Medicine, Seoul 05505, Republic of Korea. [8]Center for Augmented Safety Systems with Intelligence, Sensing and Tracking (ASSIST), Korea Institute of Science and Technology (KIST), Seoul 02792, Republic of Korea. [9]These authors contributed equally: Eunyoung Jeon, Bonhan Koo, Suyeon Kim. ✉e-mail: rhokyun@hanyang.ac.kr; shinyongno1@yonsei.ac.kr; joonseoklee@hanyang.ac.kr

microfluidic systems handle larger sample volumes and efficiently transport target molecules, such as cells, bacteria, viruses, and NAs, toward the surface[13–18]. Nevertheless, when considering the crucial liquid-solid interface, microfluidic chips with smooth and flat surfaces exhibit limitations in enhancing binding efficiency due to a restricted binding site and a non-slip boundary condition.

One representative way to tackle this issue is implementing micro-mixers to promote convective mass transfer toward the surface[19–21]. While the micro-mixing helps to deliver targets closer to the surface and shorten the necessary diffusion length[22], it remains constrained by the non-slip boundary condition. The more direct approach is to enhance the surface area[23] and slip flow through a three-dimensional (3D) porous structure[11,24,25]. Nanofabrication techniques[26,27], such as patterning[25], lithography[28], and laser printing[29], have been employed to create these porous structures, but these are technologically intensive, highly sophisticated, and time-consuming, limiting their ability to meet the growing demand for microfluidic systems as sample preparation chips. For instance, Zhang et al. (ref. 25) developed a 3D-nanopatterned microfluidic chip for exosome detection. The chip's nanoporous domain increased the surface area, facilitated drainage flow, and reduced hydrodynamic resistance near the surface, improving entrapment efficiency. However, due to the chip's small size (ranging from micrometers to centimeters) and the nanoporous domain occupying 30–70% of the total channel height, the chip has limited throughput. Therefore, additional sample preparation processes such as ultracentrifugation are required to isolate and concentrate the samples.

We introduce a simple, cost-effective fabrication method for a large-scale sample preparation chip, assembled using two polyethylene terephthalate (PET) films and double-sided tape engraved with a microchannel. The chip incorporates thin and uniform nanostructures with both small and large pores, referred to as a biporous silica nanofilm (BSNF). In computational simulations and particle proximity tests, these unique nanostructures significantly enhance the surface area and surface slip flow while inducing nanovortex in each nanopore, improving the enrichment/isolation of pathogens and NAs. By incorporating a BSNFs-embedded sample preparation chip (BSNFs-chip) into a PCR-based assay, we achieved a 100-fold reduction in the limit of detection (LOD) compared to the LOD obtained using conventional methods. Furthermore, the integration of the enriched samples from the BSNFs-chip with the luminescence resonance energy transfer (LRET) assay demonstrates its potential for rapid and sensitive detection of severe acute respiratory syndrome coronavirus 2 (SARS-CoV-2) RNA in clinical samples.

## Results

### Working principle, design, and fabrication of the BSNFs-chip
Figure 1 illustrates the workflow involved in preparing the BSNFs-chip for pathogen and NA enrichment/isolation. The BSNFs were synthesized on a large-scale commercial PET film (10 cm × 7 cm), serving as the top and bottom substrates for the sample preparation chip. The two-layered BSNF schematic synthesis is divided into four steps: (i) synthesis of the initial silica layer with small pores; (ii) removal of the micellar templates; (iii) synthesis of the subsequent silica layer with large pores; and (iv) removal of the micellar templates. The pH-dependent interactions among the micellar components influence micelle aggregation[30,31], leading to the formation of small pores in the porous silica nanofilm (PSNF) and large pores in the BSNF (Supplementary Fig. 1). The BSNFs-chip is assembled by connecting two BSNF-synthesized PET films with a microchannel-engraved double-sided tape. The BSNF's surface within the microfluidic channel is functionalized with 3-aminopropyl(diethoxy)methylsilane (APDMS) to facilitate pathogen and NA binding (Supplementary Fig. 2). Subsequently, the BSNF-chip is loaded with a clinical sample and adipic acid dihydrazide (ADH) as a cross-linking agent[32,33]. The amine-functionalized surface of

the BSNF (BSNF-NH₂) captures pathogens and NAs via electrostatic coupling and covalent binding with ADH cross-links. Finally, a concentrated sample is obtained by introducing an elution buffer into the microchannel.

### High surface area and slip flow properties of the BSNF
We utilized scanning electron microscopy (SEM) to characterize the biporous nanostructures on the flat Si wafer and PET film substrates (Fig. 2a and Supplementary Fig. 3). The top-view SEM images and the histogram of the Feret diameter distribution reveal that the first layer, a PSNF[34], is a highly uniform film with a mean Feret diameter of ~59.45 nm (Fig. 2b and Supplementary Fig. 4). Two-layered BSNF was formed through micelle aggregation and silica growth during the second layer's synthesis[35,36]. The mean Feret diameter of the second layer of the BSNF was ~230.9 nm, larger than that of the PSNF's first layer. The small-angle X-ray scattering (SAXS) profile of the BSNF displays a scattering peak at ~0.45 nm⁻¹, indicating the PSNF's presence, while a decrease in intensity is attributed to the larger pores' increased distance[37] (Supplementary Fig. 5). The cross-view SEM image of the BSNF reveals a total height of ~400 nm, with a distinct boundary between the first (height approximately 145 nm) and second layers (height approximately 255 nm) (Fig. 2c and Supplementary Fig. 6). The two BSNFs (top and bottom layers) account for a total height of approximately 800 nm within the microchannel. Notably, the BSNF represents a thin coating, occupying only 0.27% of the total microchannel height of 300 μm.

Virtual models that simulate real pores were developed to investigate the nanostructures' surface area (Fig. 2d–f)[38]. In comparison to the flat surface, the simulated surface area of the PSNF rose significantly by 1244%, while that of the BSNF increased by 1863% (Fig. 2g). Figure 2h shows a photograph of the BSNF synthesized on a large-area transparent PET film and Fig. 2i presents a picture of the BSNFs-chip fabricated with BSNFs. We evaluated the capture efficiency of the BSNFs-chip with positively charged surfaces by injecting negatively charged quantum dots (QDs) (Supplementary Fig. 7). The mean fluorescence intensity of the PSNFs-embedded sample preparation chip (PSNFs-chip) exhibited a higher value compared to the existing sample preparation chip with flat surfaces (Flat-chip), while the BSNFs-chip demonstrated an even greater intensity than the PSNFs-chip, thereby indicating its superior capture capacity. These results confirm that the increased surface areas of the PSNF and BSNF, compared to the bare PET film's flat surface, are due to the nanostructures.

We conducted a numerical simulation of three virtual models (Flat, PSNF, and BSNF) to scrutinize fluid flows and evaluate the BSNFs-chips' enhanced mass transfer (Supplementary Fig. 8 and Supplementary Table 1, refer to Methods for detailed simulation setup)[39]. This enhancement was achieved by (i) eliminating non-slip conditions on the nanostructure; and (ii) generating vortical flows within the nanostructures (Fig. 3 and Supplementary Movies 1 and 2). As could be intuitively expected, the introduction of porous structures facilitated fluid flow on and within the nanostructure. However, there was no flow at the no-slip bottom on the bare PET (Supplementary Fig. 9 and Supplementary Movie 3). A nanoscale vortex was formed in each pore due to the tangential slip flow in the PSNF and BSNF models. These vortices drive fluid flow toward the nanostructures, thus enhancing mass transport from the bulk fluid and the frequency of particle-surface collisions (i.e., binding efficiency).

To quantify the slip velocity and the strength of the vortical flows within the nanostructures, we estimated the vorticity strength and average velocity profile in the Flat PET, PSNF, and BSNF (Fig. 3g–i). The velocity profiles revealed that the PSNF demonstrates considerable slip velocity (x-intercept, i.e., the velocity at the liquid-structure interface) and slip length (y-intercept) ($3.28 \times 10^{-9}$ m s⁻¹ and 55.6 nm)[40]. However, the BSNF displays a 10-fold higher slip property ($2.67 \times 10^{-8}$ m s⁻¹ and 300 nm) (Fig. 3i). The robust slip flow on the BSNF also

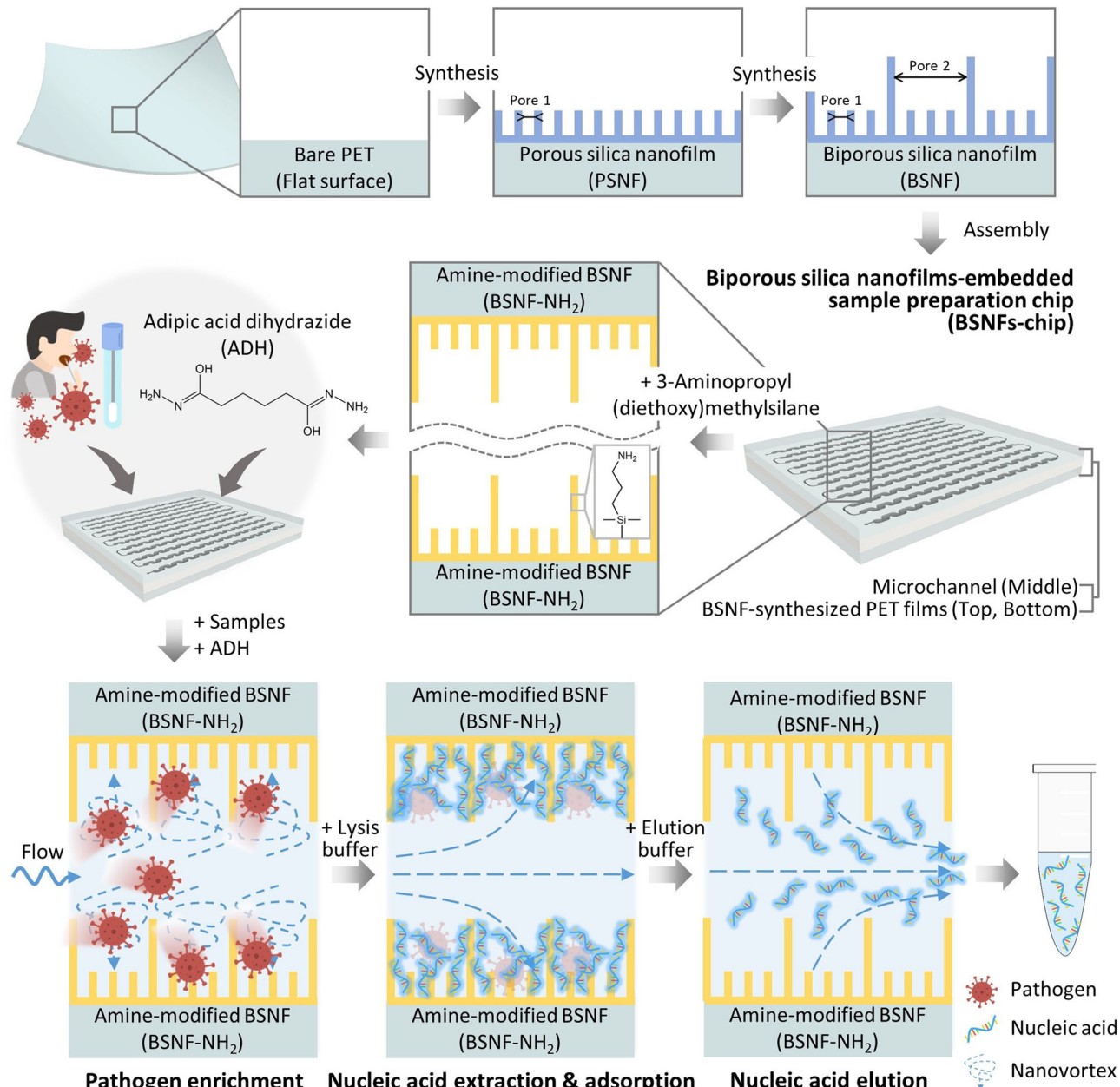

**Fig. 1 | Schematic drawings of the BSNFs-chip design and the sample preparation process.** The polyethylene terephthalate (PET) film serves as a substrate for nanofilm synthesis and the construction of a sample preparation chip. The biporous silica nanofilm (BSNF) is formed by synthesizing a small-pore porous silica nanofilm (PSNF) on the PET film, followed by sequential synthesis of a large-pore silica nanofilm on the PSNF. The BSNFs-chip is fabricated by assembling two PET films, which have been synthesized with the BSNF, into a microchannel-engraved tape. The BSNF surfaces corresponding to the microchannel are modified with amine-functional groups for sample binding. When samples and ADH are injected into the BSNFs-chip, the amine-functional groups of the BSNF surface bind ADH and pathogens, resulting in pathogen enrichment. This enrichment process is more efficient due to the high surface area and nanovortex of the BSNF. A lysis buffer is added to the microchannel to extract nucleic acids from the enriched pathogens, and the enriched nucleic acid sample is collected using an elution buffer.

prompts intensive vortical flows within the nanostructures, with vorticity strengths of $4.0 \times 10^{-3}$ $s^{-1}$ (BSNF) and $2.2 \times 10^{-3}$ $s^{-1}$ (PSNF) (0 for Flat PET). Notably, Fig. 3j demonstrates that the vorticity strength predicts the results of the two capture tests (the QD capture test in Supplementary Fig. 7 and the pathogen capture test in Fig. 4b). Furthermore, if we evaluate the velocity profile's slope on the surface, which is proportional to the wall shear stress, we can also estimate the nanostructures' slip quality. The BSNF and PSNF have much smaller slopes (-1) than other nanostructures presented in previous studies (e.g., approximately 60)[25], likely resulting from the nanostructures containing large voids with minimal solid partitions.

Lastly, we executed a particle proximity test to directly visualize the slip flows on the BSNF (Fig. 3k and Supplementary Movie. 4). When we propel the 10 μm fluorescent particles in the solution toward a vertically aligned solid surface under perfect slip conditions with an ideally inviscid flow, the streamline (or particle trajectories) shows a two-dimensional stagnation flow with a single stagnation point at the center of the solid surface. While microscale particles cannot trace fluid flows in or right above the nanoscale porous structures, they can draw fluid streamlines and the boundary layer near the surface qualitatively, which will be changed according to the slip length and/or permeability of the surface. In this scenario, the no-slip wall with zero

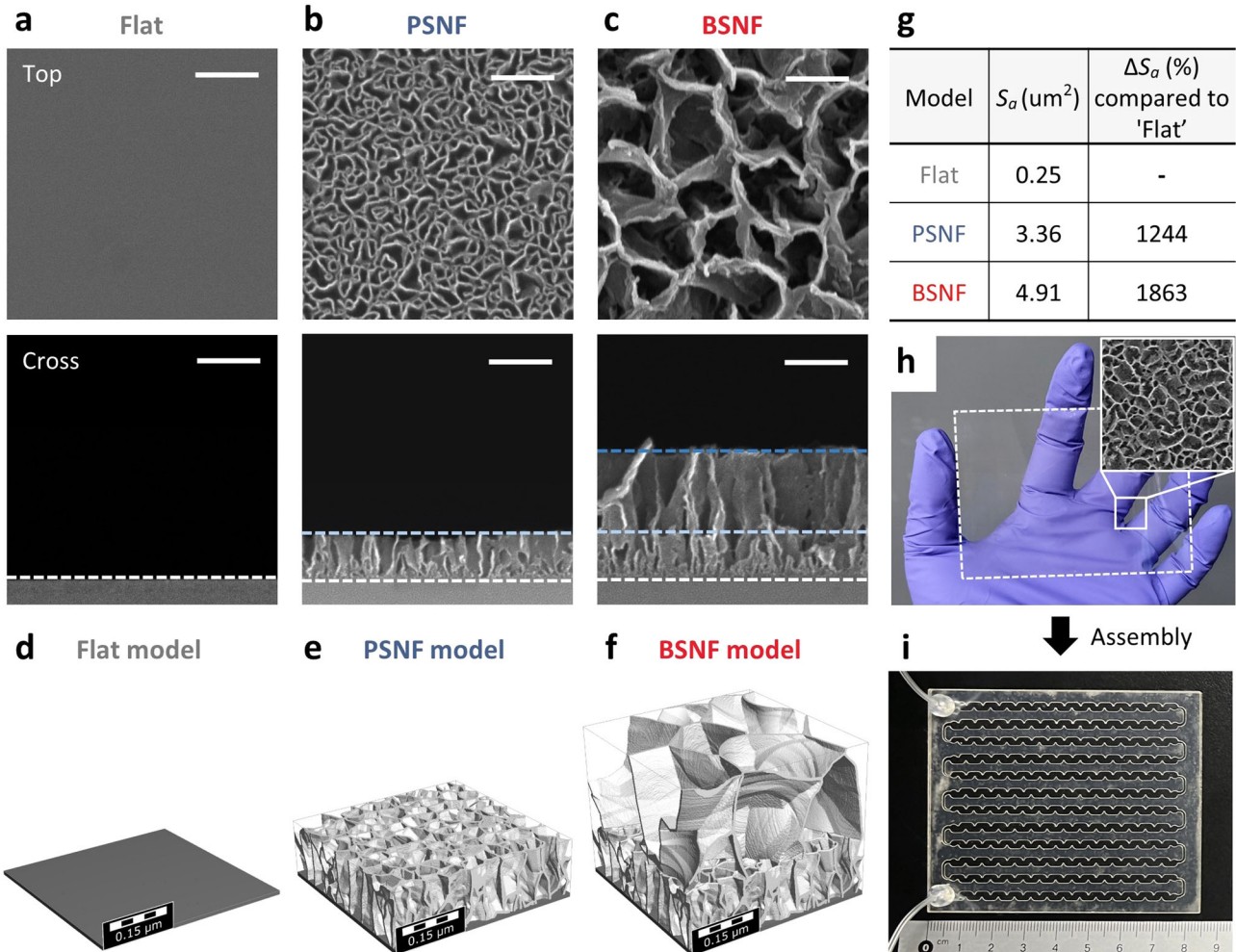

| Model | $S_a$ (um$^2$) | $\Delta S_a$ (%) compared to 'Flat' |
|---|---|---|
| Flat | 0.25 | - |
| PSNF | 3.36 | 1244 |
| BSNF | 4.91 | 1863 |

**Fig. 2 | Large-scale and uniform synthesis of the BSNF with high surface area. a–c** Top-view (top) and cross-sectional (bottom) SEM images of **a** flat surface (Flat), **b** PSNF, and **c** BSNF on Si wafer. Data in (**a–c**) are representative of *n* = 3 independent experiments. Scale bars in (**a–c**), 200 nm. **d–f** 3D virtual model images of **d** Flat, **e** PSNF, and **f** BSNF. **g** Comparison table of simulated surface area ($S_a$) of different virtual models. **h** Photograph of a large-scale BSNF-synthesized PET film. Inset: high-magnification SEM image. **i** Photograph of the BSNFs-chip. Colors in gray, blue, and red represent Flat, PSNF, and BSNF, respectively.

flow velocity on the bare PET prevents the particles from flowing near the membrane, which results in a fluorescent vacancy near the stagnation point and the surface (the distance of the closest particle trajectory from the stagnation point is 57.7 μm, Fig. 3k)[24]. However, the BSNF, with its enhanced slip flow, can decrease this distance to 22.5 μm (~61% reduction), indicating that the targets can approach and be captured by the BSNF more easily.

### Assessment of BSNFs-chip for pathogen and NA enrichment/isolation

We conducted experiments to enrich and isolate pathogens and NAs using the BSNFs-chips, leveraging the enhanced capture efficiency of the BSNF described above. Supplementary Figure 10 visually depicts the pathogen and NA enrichment/isolation processes using the BSNFs-chip. Within the microchannel, negatively charged pathogens and NAs were captured and enriched on the chip's surface using ADH and subsequently separated using a high pH (pH 10–11) elution buffer (Supplementary Fig. 11). By employing ADH as a non-chaotropic and homobifunctional hydrazide cross-linking agent, the BSNFs-chip eliminates the requirement for chaotropic agents, thereby preventing NA degradation. We initially assessed the pathogen and NA enrichment efficiencies of the sample preparation chips quantitative reverse-transcription PCR (qRT-PCR), comparing these results with those obtained using a conventional NA isolation method. Figure 4a

presents a schematic overview of the conventional method (isolation without enrichment), the 2-Step method (pathogen enrichment followed by NA isolation without NA enrichment), and the 1-Step method (sequential enrichment/isolation of pathogens and NAs). The 2-Step method demonstrated lower cycle threshold (Ct) values than the conventional method, indicating efficient pathogen enrichment by the chips (Fig. 4b). Moreover, the 1-Step method, which employs the NA enrichment capability of the sample preparation chips, displayed lower Ct values than the 2-Step method. Additionally, in both the 2-Step and 1-Step methods, the Ct values showed a decreasing trend in the order of the Flat-, PSNFs-, and BSNFs-chip, correlating with the enhanced surface binding efficiency. Notably, the BSNFs-chip, utilizing the 1-Step method, exhibited superior capture efficiency and a nearly identical absolute Ct value (30.18).

To validate the improvement in NA detection sensitivity achieved by the sample preparation chips, we evaluated the LOD for the 1-Step method using serial dilutions of HCT116 cells for cell, genomic DNA, and RNA enrichment, as well as SARS-CoV-2 culture fluid for virus and viral RNA enrichment (Supplementary Tables 2–5). The sample preparation chips effectively concentrated each cell and virus sample from 1 ml to a final volume of 100 μl. These features resulted in lower Ct values for all samples, both genomic NAs and viral RNA, processed via the Flat-chips, indicating a 10-fold lower LOD of the Flat-chip compared to the conventional method (Fig. 4c–e). In addition, the PSNFs- and

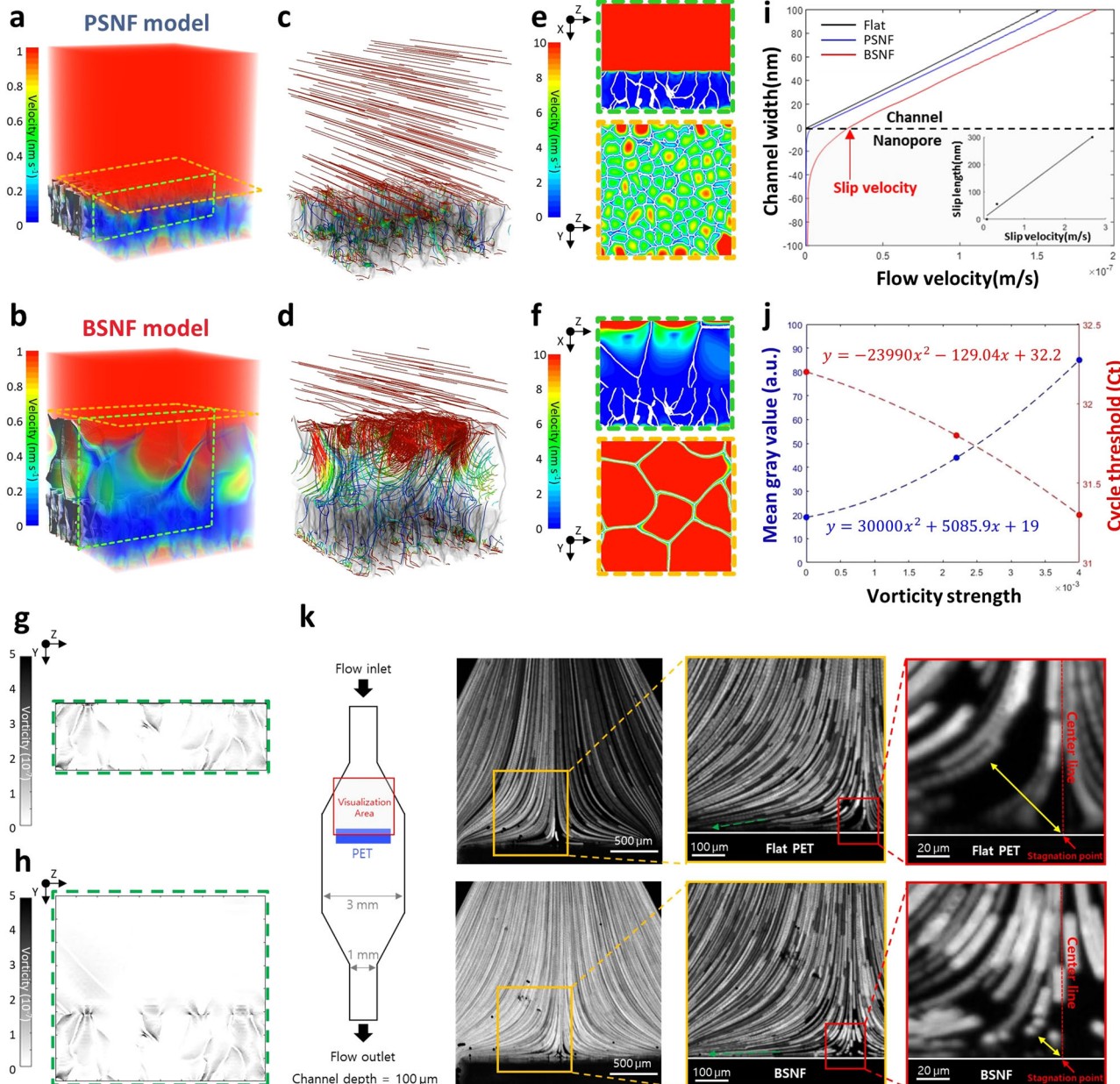

**Fig. 3 | Dynamic flow profiles of the BNSF. a–d** 3D and **e–h** 2D plane images of flow velocity field, streamlines, and vorticity field in the virtual models under a given average flow velocity ($7.7 \times 10^{-7}$ m s$^{-1}$); (**a, c, e, g**) PSNF and (**b, d, f, h**) BSNF (see "Methods" for detail simulation setups). The vorticity strength is calculated as the sum of the absolute value of each velocity gradient($|\partial v/\partial z|+|\partial w/\partial y|$), instead of the curl of the velocity because Geodict only provides the absolute values of the velocity. **i** Simulated average velocity profiles on Flat, PSNF, and BSNF. The slip velocity (x-intercept, i.e., the velocity at the liquid-structure interface) and the slip length (y-intercept) are 0/0 (Flat), $3.28 \times 10^{-9}$ m s$^{-1}$, 55.6 nm (PSNF), and $2.67 \times$ $10^{-8}$ m s$^{-1}$, 300 nm (BSNF). The slip velocity and the slip length show linear proportionality (inset graph). **j** Correlation between the vorticity strength and two capture efficiency tests (quantum dots (QDs) capture (Mean gray value in Supplementary Fig. 7) and pathogen capture (Ct in Fig. 4b). a.u., arbitrary units. **k** The schematic of the particle proximity test to confirm slip effects of BSNF and the visualization result of particle flow paths (10 μm polystyrene beads, 20 μl min$^{-1}$ of the flow rate). The large slip velocity of BSNF allows particles to migrate closer to the surface; the distance of the closest particle from the stagnation point is 57.7 μm (Flat) and 22.5 μm (BSNF). Source data are provided as a Source data file.

BSNFs-chips not only concentrated but also enriched the pathogens/ NAs through nanostructures, resulting in lower Ct values and LOD than the Flat-chip. As depicted in Fig. 4c, for the DNA samples, both BSNFs- and PSNFs-chip showed 10-fold lower LOD ($1 \times 10^1$ cell ml$^{-1}$) than the Flat-chip and a 100-fold lower LOD than the conventional method. For RNA samples (both genomic RNA and viral RNA), the BSNFs-chip demonstrated a 10-fold higher sensitivity (LOD of $1 \times 10^1$ cell ml$^{-1}$ for genomic RNA and $0.96 \times 10^0$ PFU ml$^{-1}$ for viral RNA) than the Flat- and PSNFs-chips, and a 100-fold higher sensitivity than the conventional method (Fig. 4d, e). These differences could be attributed to factors

such as the disparate chemical stability between the robust double-helix structure of DNA and the less stable single-stranded structure of RNA, differential electrostatic coupling to the positively charged groups (RNA carries a weaker negative charge than DNA), and the additional reverse transcription step required for RNA in qRT-PCR[41–46]. The stability of the BSNFs-chip was confirmed over a 2-week period immediately following chip fabrication through SEM imaging and FT-IR spectroscopy (Supplementary Fig. 12). The structural stability of the BSNFs-chip was also confirmed even after being used for pathogen and NA enrichment/isolation (Supplementary Fig. 13). This robustness

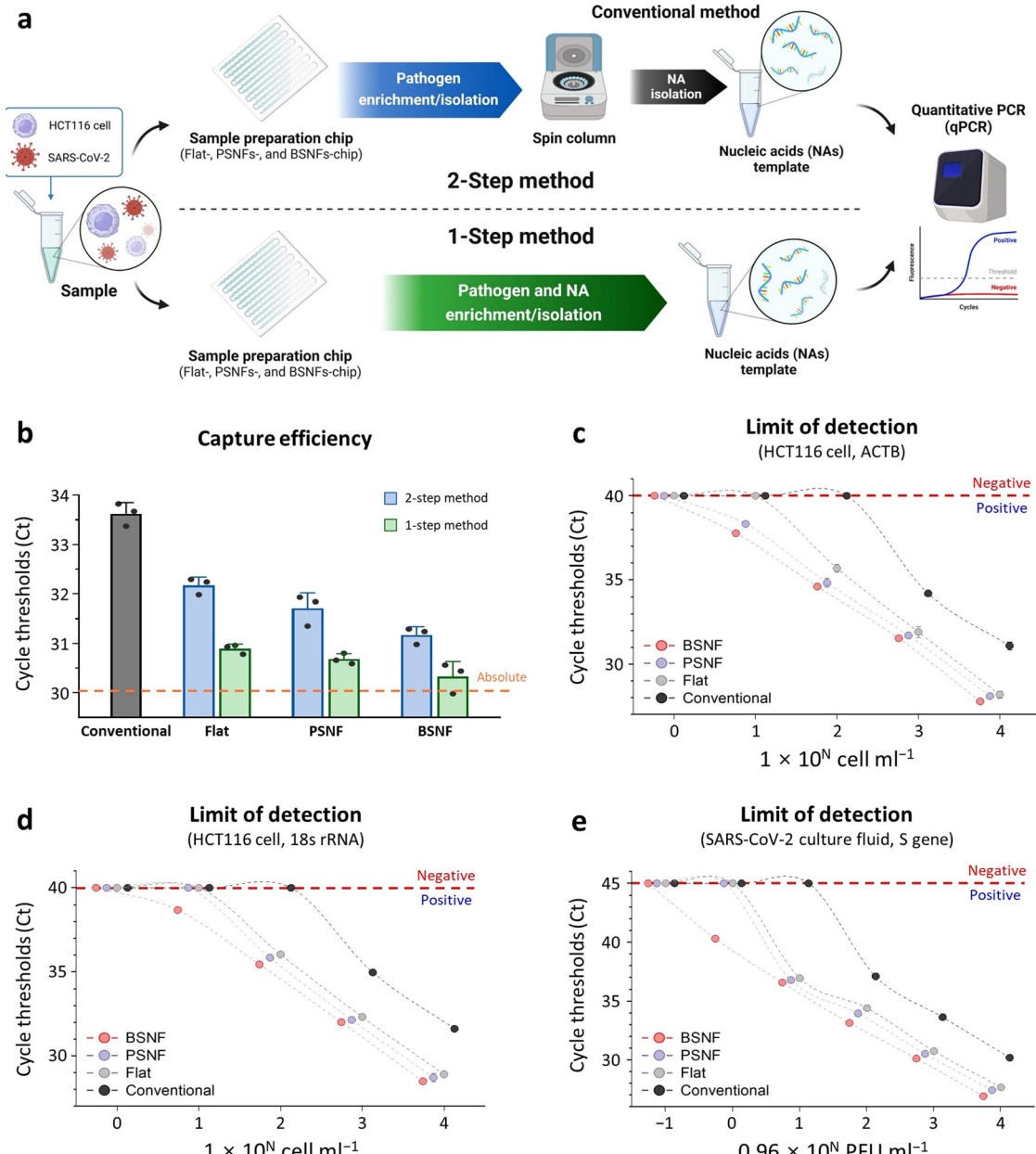

**Fig. 4 | Evaluation of pathogen and NA enrichment/isolation on a BSNFs-chip.**
**a** Schematic representation of strategies to analysis pathogen and nucleic acid (NA) enrichment/isolation efficiencies. The validation of these strategies was achieved through two methods; The 2-Step method involved initial concentration and isolation of pathogens using Flat-chip, PSNFs-chip, and BSNFs-chip, followed by conventional isolation of NAs. The 1-Step method allowed simultaneous pathogen and NA enrichment/isolation within a single Flat-chip, PSNFs-chip, and BSNFs-chip. Figure created with BioRender.com. **b** Comparative analysis of pathogen and NA enrichment/isolation capture efficiency between the 1-Step and 2-Step methods with a conventional NA isolation method. The orange dashed line represents the absolute mean value at 30.18. For each method, $n = 3$ biologically independent experiments were conducted. Data are presented as mean values ± SD. **c**–**e** Limit of detection (LOD) results employing the 1-Step method for DNA targeting the ACTB gene from HCT116 cells ($n = 3$ for the conventional method and $n = 4$ for Flat, PSNF, and BSNF in biologically independent experiments), for RNA targeting the 18 s rRNA gene from HCT116 cells ($n = 3$ for the conventional method and $n = 4$ for Flat, PSNF, and BSNF in biologically independent experiments), and for viral RNA targeting the S gene from SARS-CoV-2 ($n = 3$ biologically independent experiments per group). Data are presented as mean values ± SD. Source data are provided as a Source data file.

makes the BSNFs-chip suitable for both disposable and reusable applications. Notably, the chip demonstrated consistent performance across multiple uses, with Ct values of 27.78 ± 0.13 for the first use, 27.3 ± 0.35 for the second use, and 27.03 ± 0.30 for the third use, indicating high reproducibility. These results further establish the BSNFs-chip as a promising sample preparation platform for the accurate diagnosis of infectious diseases, highlighting its potential as a reliable tool in the sensitive and precise diagnosis of infectious diseases.

## PCR-free detection of RNA from clinical samples from SARS-CoV-2 patients and healthy

To address the challenges of complexity and time-consuming qRT-PCR for SARS-CoV-2 detection[47,48], we devised a nanoparticle-based LRET assay as a rapid and simplified alternative. This assay utilizes the principle of distance-dependent non-radiative energy transfer between the energy donor and acceptor, allowing for target identification without the need for complicated steps[49,50]. We designed a strategy that combines a 1-Step sample preparation chip and a LRET

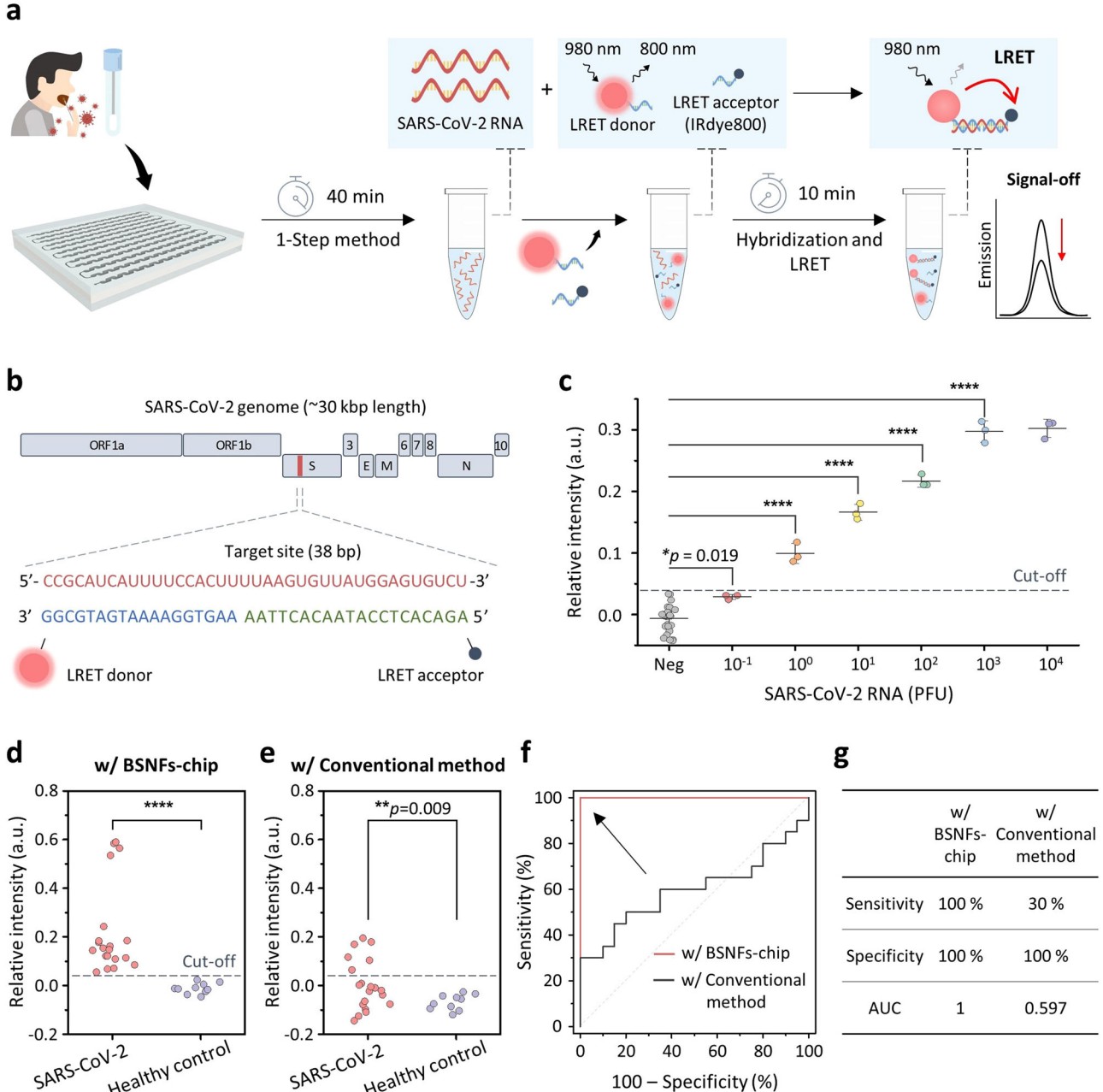

**Fig. 5 | High-sensitivity PCR-free detection of SARS-CoV-2 RNA extracted from BSNFs-chip. a** Workflow of the luminescence resonance energy transfer (LRET) assay for SARS-CoV-2 RNA collected using BSNFs-chip. **b** Schematic diagram of the full-length SARS-CoV-2 genome map showing the target gene site for the LRET assay. **c** Analytical sensitivity of the LRET assay for serial diluted SARS-CoV-2 RNA. a.u., arbitrary units. Data are expressed as mean ± SD ($n = 3$ independent experiments). $p$ values: $p = 0.019$ ($10^{-1}$ PFU), $p = 1.37E-7$ ($10^0$ PFU), $p = 3.49E-12$ ($10^1$ PFU), $p = 5.02E-15$ ($10^2$ PFU), $p = 1.42E-18$ ($10^3$ PFU), $p = 8.18E-19$ ($10^4$ PFU). **d**–**g** Diagnostic accuracy of the LRET-based detection towards clinical samples, including 20 SARS-CoV-2 positive samples and 10 negative samples. (**d**, **e**) The LRET assay integrated

with the BSNFs-chip shows significantly higher sensitivity ($p = 0.000019$) than with a conventional extraction method ($p = 0.009$). Data are expressed as mean ($n = 3$ independent experiments). **f**, **g** The LRET assay integrated with the BSNFs-chip has significantly higher accuracy (sensitivity of 100% and area under the ROC curve (AUC) of 1) than that integrated with the conventional method (sensitivity of 30% and AUC of 0.597). $p$ values in (**c**–**e**) were determined by a two-tailed unpaired t-test; $*p < 0.05$, $**p < 0.01$, $***p < 0.001$ and $****p < 0.0001$. The cut-off value (0.0395) was determined by applying optimal combinations of clinical sensitivity and specificity from receiver operator characteristic (ROC) curve based on the Youden index point. Source data are provided as a Source data file.

assay, leading to a substantial reduction in the sample-to-answer time from several hours to under an hour for SARS-CoV-2 diagnosis. This strategy begins with the enrichment/isolation of pathogens and NAs using the BSNFs-chip (approximately 40 min), which is followed by the identification of enriched target RNAs using the LRET assay (approximately 10 min), and the entire process from sample collection to obtaining results for SARS-CoV-2 detection is completed within 50 min (Fig. 5a). The LRET system consists of core/active-shell/shell

lanthanide-doped nanoparticles (LRET donor, 800 nm luminescence under 980 nm excitation) and IRdye800 (LRET acceptor, with strong and broad absorption ranging from 600 to 850 nm) (Supplementary Fig. 14 and Supplementary Note 1)[51,52]. The strands complementary to the target site (38 base pair (bp) length), which forms part of the spike (S) gene strand[53–55], were divided into two half-sequences, which were then modified on the surfaces of the LRET donor (18 bp DNA) and acceptor (20 bp DNA) (Fig. 5b, Supplementary Fig. 15 and

Supplementary Table 6). In the presence of SARS-CoV-2 RNA, oligo-nucleotide hybridization occurs between complementary pairs, bringing the LRET donor and acceptor into close proximity. This property allows the acceptor near the donor to absorb the 800 nm emission of the donor via LRET under 980 nm excitation. The relative intensities increase with the rising concentrations of the target RNA, which are calculated as the ratio of the decreased luminescence intensities of the LRET donor. Furthermore, the DNA oligos on the surface of the LRET donor and acceptor could hybridize specifically to SARS-CoV-2 RNA with no obvious cross-reactivity to other non-target NAs (Supplementary Fig. 16a). We validated the specificity of the LRET assay using 1 pM of non-target sequences of other common contagious respiratory viruses. The LRET assay successfully distinguished SARS-CoV-2 from human coronavirus OC43 (hCoV-OC43), hCoV-NL63, hCoV-229E, and influenza A virus (IAV) (Supplementary Fig. 16b).

We evaluated the analytical sensitivity of the LRET assay using 10-fold serial dilutions of full-length SARS-CoV-2 genomic RNA (approximately 30 kbp), ranging from $10^{-1}$ to $10^4$ PFU. The relative intensities displayed a linear relationship with the logarithmic concentration of SARS-CoV-2 RNA ranging from $10^{-1}$ to $10^{-3}$ PFU, with a high correlation coefficient ($R^2$) of 0.998 (Fig. 5c and Supplementary Fig. 16c, d). The detection results for all samples were statistically different from those of the negative control. The LOD was estimated to be $10^{-0.93}$ PFU based on the equation derived from the standard curve (Supplementary Fig. 16d). After adding $10^3$ PFU SARS-CoV-2 RNA, the 800 nm luminescence lifetime was quenched from 293 µs to 265 µs, indicating the occurrence of non-radiative energy transfer from donor to acceptor via hybridization between complementary pairs[56] (Supplementary Fig. 16e). Finally, we tested the clinical applicability of the LRET assay in conjunction with the BSNFs-chip using 30 nasopharyngeal (NP) swab samples. For patient samples identified as positive by qRT-PCR, the LRET assay in combination with the BSNFs-chip also yielded positive results with $p$-values < 0.0001. However, the LRET assay combined with the conventional extraction method resulted in substantial false-negative results (Fig. 5d, e and Supplementary Figs. 17 and 18). We conducted a receiver operator characteristic (ROC) analysis to examine the diagnostic accuracy, wherein the LRET assay with the BSNFs-chip successfully discriminated between SARS-CoV-2-positive and -negative samples with a sensitivity of 100.0% and an area under the ROC curve (AUC) value of 1 (Fig. 5f, g). In contrast, the LRET assay combined with the conventional method exhibited low sensitivity of 30.0% and an AUC value of 0.597. These results indicate the potential for PCR-free detection of infectious diseases through the integration of a rapid diagnostic system with BSNFs-chip-based sample preparation for pathogen/NA enrichment.

## Discussion

We developed the BSNFs-chip, a sample preparation chip with the biporous nanostructures that synergize the benefits of increased surface area and surface slip flow. The BSNFs-chip, made with a PET film costing under $1, presents a cost-effective strategy, and the enclosed system reduces the risk of cross-contamination. Our thorough investigation of the underlying mechanisms through simulations and experiments demonstrates the contribution of small and large pores in augmenting BSNFs' surface area and slip flow that even induce nanovortex enabling better NA isolation. Based on the high binding probability of the BSNFs-chip validated through simulations and experiments, we assessed its effectiveness in enriching pathogens and NAs using PCR-based methods. The 100-fold lower LOD of the BSNFs-chip compared to conventional extraction methods is crucial for efficient infectious disease detection, especially at low pathogen level. While LRET-based detection alone has lower sensitivity than PCR-based methods, the integration of the 1-Step sample enrichment system and LRET assay allowed for successful discrimination of SARS-

CoV-2-positive and -negative samples with a sensitivity of 100.0%. Our PCR-free sensing approach, which combining a well-designed sample preparation system with a detection system, paves the way for efficient and sensitive diagnosis of infectious diseases. Future investigations will focus on extending the applicability of these nanomaterials through exploring the micelle aggregation mechanism and precise pore control, including the design of customized nanostructures, while simultaneously aiming to enhance the integration of the BSNFs-chip with the LRET assay to expand their combined utility beyond SARS-CoV-2 detection to a broader range of pathogens. We anticipate that this approach will play an impactful role in advancing the field of rapid and sensitive diagnostic methods for infectious diseases in the years to come.

## Methods

### Synthesis of PSNF and BSNF

2.992 g of triethanolamine (Sigma-Aldrich, 90279) was dissolved in 1.045 L of distilled water at 80 °C and 100 µm thin PET films (Kemafoil hydrophilic film, HNW-100, COVEME, Italy, 10 cm by 7 cm) were immersed in the solution. After 30 min, 61.16 ml of cetyltrimethylammonium chloride (CTAC) (Sigma-Aldrich, 292737) and 12.496 g of sodium salicylate (NaSal) (Sigma-Aldrich, S3007) were added under vigorous stirring to form micelle templates[36]. Subsequently, 110 ml of tetraethyl orthosilicate (TEOS) (Sigma-Aldrich, 86578) was added into the solution and stirred at 80 °C for 3 h. The PSNF-synthesized PET films were washed three times with ethanol and water and dried for further use.

The PSNF with small pores around 60 nm was employed not only as the binding site with high surface area, but also as the seed layer for the synthesis of BSNF. A mixture of 1.045 L distilled water, 61.16 ml of CTAC and 12.496 g of NaSal were stirred vigorously to form giant micelle templates at 80 °C for 2 h. PSNF-synthesized PET films were immersed in the solution for 2 h under stirring. Then, 110 ml of TEOS was added and stirred for 2 h. Finally, BSNF-synthesized PET films were washed three times with ethanol and water and stored in dry.

### Design and fabrication of the BSNFs-chip

Microfluidic chips, which include the Flat-chip, PSNFs-chip, and (BSNFs-chip, are structured as three-layered devices. Each layer is composed of either a 100 µm flat PET film, PSNF, or BSNF for the top and bottom layers. In the center, a layer features double-sided tape structures, which are 300 µm in height and arranged in a spiral shape. These structures connect the top and bottom films and manage the circulation of fluid within the microchannels. The overall external height of the microfluidic chip reaches 500 µm, while the internal microchannels maintain a height of 300 µm. The designs for the film and the double-sided tape were drafted using AutoCAD 2019 software (Autodesk, Inc., San Rafael, CA, USA). Both the film and the double-sided tape structures were shaped into rectangles measuring 70.16 mm by 85 mm. Specifically, the double-sided tape structures were designed as a repeated sequence of 12 microfluidic channels, each composed of 13–14 half-oval shapes with a diameter of 4.5 mm. This design enabled a serpentine flow and the injection of ~650 µl of liquid. Following the design process, the layouts were cut into their desired shapes using the VLS3.50 laser machine (Universal Laser Systems, Scottsdale, AZ, USA)[33].

Before usage, the microfluidic chips were assembled, and the internal channels were treated to promote amine group functionality. The initial step involved subjecting the films to $O_2$ plasma treatment for surface cleansing and alteration, which led to the production of hydroxyl groups (OH) to enhance hydrophilicity and chemical reactivity. Once treated, the layers were precisely aligned and fastened together. Acrylic adaptors were then affixed to the inlet and outlet of the microfluidic chip's upper layer to facilitate sample injection and

expulsion. Tygon tubing was linked to the adaptor openings using epoxy, thus ensuring a robust connection for fluid movement. Following this, the inner surface of the microfluidic chips was functionalized by introducing a 2% solution of 3-aminopropyl(diethoxy) methylsilane (APDMS; Cat no. 371890, Sigma-Aldrich, St. Louis, MO, USA) into ultra-pure deionized water (DW). Subsequently, the microfluidic chips were incubated at 65 °C for 60 min and then washed thrice with 1 ml of DW each time to eliminate any residual silane. After the fabrication process was concluded, the amine group functionalized Flat-chip, PSNFs-chip, and BSNFs-chip were stored at ambient temperature until needed for further use.

## Simulation

Computational modeling and simulations were carried out via the commercial solver of the GeoDict® 2023 software package (Math2-Market GmbH, Germany). GeoDict is a voxel-based tool and is used for prediction of the surface area and flow performance of the simulated virtual models. Virtual models were generated with the FiberGeo module and the Sinter & Crystallization algorithm of the GrainGeo module. The pore diameter and the thickness of pore wall were constructed by the approximate values of the experimental values. Domains discretized with a voxel length of 1 nm were made to be $500 \times 500$ nm² (x–z plane), and 3000 nm (y-axis) in flat, 3145 nm (y-axis) in PSNF, and 3400 (y-axis) nm in BSNF, respectively. Then, MatDict module was used to calculate the surface area of the simulated virtual models.

A FlowDict module was used to calculate the flow profiles for the simulated virtual models. The flow of water was obtained from the Stoke's equation considering at a given flow condition. The water flow was passed in z-direction, and 100 µl min⁻¹ of flow rate and 21.66 cm² of area (inner surface area of microchannel) were imposed as the experimental setup. The periodic boundary condition was employed in the peripheral region to model the infinite periodic domains.

## Particle proximity test

To confirm the slip effect of BSNF, we conducted a particle proximity test through the visualization of particle flow paths. A micro-scale channel (3 mm width, 0.1 mm height) was fabricated using transparent polydimethylsiloxane (PDMS) for visualization. An inlet and outlet were placed on both sides of the channel, and a 2 mm wide PET film was placed at the middle of the channel vertically (Fig. 3k). Then, an aqueous solution containing 10 µm fluorescent polystyrene beads (Phosphorex, Inc., USA) was injected into the channel at a flow rate of 20 µl min⁻¹ using a syringe pump (Fusion 200, Chemyx Inc., USA). The trajectories of the particles were captured at every 0.05 s using an inverted microscopy (IX-71, Olympus Co., Japan) and EMCCD (ImagEMx2, Hamamatsu Co., Japan). The particle flow paths were visualized by stacking 50 frames of images (i.e., during 2.5 s).

## Material characterization

SEM images were obtained using a ZEISS Sigma 300 field-emission SEM (FE-SEM) (ZEISS, Germany) at the Center for Polymers and Composite Materials, Hanyang University, Korea. The zeta potential was measured by a Malvern Zetasizer (Malvern, UK). Feret diameter and mean gray value were analyzed in ImageJ software. SAXS measurement was performed using a Xeuss 2.0 (Xenocs, France) with a Cu Kα radiation ($\lambda = 0.154$ nm) and a sample-to-detector distance of 1500 mm. Fluorescence images were observed under fluorescence imaging system (EVOS M7000 Imaging System, Thermo Fisher Scientific, USA) using Qdot™ ITK™ Carboxyl Quantum Dots (ThermoFisher Scientific, USA). FT-IR spectrum was obtained using a PerkinElmer Frontier™ FT-IR spectrometer with a diamond ATR (Perkin Elmer, USA), and water contact angle was measured using pendant drop tensiometer DSA100 (Kruss, Germany).

## Conventional methods for NA isolation and detection

Conventional methods for NA isolation and detection, such as the spin column method and qPCR, were employed to evaluate and validate the BSNFs-chip for pathogen and NA enrichment/isolation, along with the LRET assay for PCR-free detection. For the NA isolation, serial diluted HCT116 cells (from $1 \times 10^4$ to $1 \times 10^0$ cell ml⁻¹) and SARS-CoV-2 culture fluid (from $0.96 \times 10^4$ to $0.96 \times 10^{-1}$ PFU ml⁻¹) were prepared in 1.5 mL microcentrifuge tubes. Genomic DNA, RNA, and viral RNA were directly extracted from 100 µl of the serial diluted samples and 100 µl of pathogens concentrated and isolated by the BSNFs-chip. This was achieved using QIAamp DNA Mini kit (Cat no. 51306, Qiagen, Hilden, Germany), QIAamp RNA Blood Mini kit (Cat no. 52304, Qiagen), and QIAamp Viral RNA Mini Kit (Cat no. 52906, Qiagen), respectively. The NAs were then eluted with 100 µL of elution buffer supplied in the kits and stored at −20 or −80 °C until further use. For NA detection, primer and probe sets were designed for each target NAs and mixed with 5 µl of eluted NAs to prepare PCR mixes. qPCR was performed for DNA of the ACTB gene (40 cycles; denaturation at 95 °C for 10 s, annealing at 60 °C for 20 s, and extension at 72 °C for 20 s) using Brilliant III Ultra-Fast SYBR Green qPCR Master Mix (Cat no. 600882, Agilent Technologies, Santa Clara, CA, USA). qRT-PCR was performed for RNA of 18 s rRNA gene (reverse transcription at 50 °C for 20 min and 40 cycles; denaturation at 95 °C for 15 s, annealing at 58 °C for 15 s, and extension at 72 °C for 30 s) using Brilliant III Ultra-Fast SYBR Green qRT-PCR Master Mix (Cat no. 600886, Agilent Technologies). qRT-PCR for viral RNA of the S gene of SARS-CoV-2 was performed (reverse transcription at 50 °C for 10 min and 45 cycles; denaturation at 95 °C for 5 s and annealing/extension at 60 °C for 30 s) using LightCycler Multiplex RNA Virus Master Mix (Cat no. 06754155001, Roche, Basel, Switzerland). The primers and probe used are listed in Supplementary Table 2. The amplified products were detected through SYBR green and Cy5 fluorescence signals using the CFX96 Real-Time PCR System (Bio-Rad, Hercules, CA, USA).

## Pathogen and NA enrichment/isolation using the BSNFs-chip

Microfluidic chips, namely Flat-chip, PSNFs-chip, and BSNFs-chip, were utilized in two different strategies: a 2-Step method focused solely on pathogen enrichment/isolation without NA enrichment/isolation, and a 1-Step method allowing simultaneous enrichment/isolation of both pathogens and NAs, as depicted in Fig. 4a. We obtained the HCT116 cell line (KCLB No. 10247) from the Korean Cell Line Bank (Seoul, Korea) and heat-inactivated SARS-CoV-2 Culture Fluid (0810590CFHI) from ZeptoMetrix (Buffalo, NY, USA). To begin with, HCT116 cells were serially diluted from $1 \times 10^4$ to $1 \times 10^0$ cells ml⁻¹, while SARS-CoV-2 culture fluid was diluted from $0.96 \times 10^4$ to $0.96 \times 10^{-1}$ PFU ml⁻¹. For the enrichment/isolation of pathogens and NAs, a volume of one milliliter from these serially diluted samples was combined with 100 mg of adipic acid dihydrazide (ADH; Cat no. 8.41689.0050, Merck Millipore, Billerica, MA, USA). The blend was then injected into the microfluidic chip's internal channel through the inlet Tygon tubing using a syringe and syringe pump at a rate of 100 µl min⁻¹. The microfluidic chip was left to incubate at room temperature for 15 min, which allowed for pathogen capture on the flat, porous, or biporous structured surface of the film. Post-incubation, debris and unreacted ADH were eliminated using an air-filled syringe, and any remaining residue was washed away with 1 ml of PBS and air.

In the 2-Step method, pathogen enrichment/isolation was achieved without NA enrichment/isolation. The concentrated pathogens were isolated using 100 µl of elution buffer with a pH of 10–11, at a flow rate of 25 µl min⁻¹. Conversely, the 1-Step method involved the simultaneous enrichment/isolation of both pathogens and NAs. The microfluidic chips were filled with pathogen lysis buffer, consisting of 20 µl Proteinase K (Cat No. 19133, Qiagen, Hilden, Germany), 100 mM pH 8.0 Tris-HCl (Cat No. 15568025, Invitrogen, Carlsbad, CA,

USA), 10 mM ethylenediaminetetraacetic acid (Cat No. AM9260G, Invitrogen), 1% sodium dodecyl sulfate (Cat No. AM9822, Invitrogen), 10% Triton X-100 (Cat No. T8787, Sigma-Aldrich), 50 mg ADH, and 10 μl RNase-free DNase (only for RNA) (Cat No. 79254, Qiagen), and incubated at 56 °C (for DNA) or room temperature (for RNA) for 15 min. This procedure facilitated pathogen lysis and NAs capture on the flat, porous, or biporous structured surface of the film. After the incubation, an air-filled syringe was used to remove pathogen debris and unused ADH from the reaction, and the remaining residues were subsequently washed with 1 ml of PBS and air. Finally, the concentrated NAs were isolated at a rate of 25 μl per minute using 100 μl of pH 10–11 elution buffer.

For the clinical use, 30 NP swab samples were collected from patients suspected of having COVID-19, of which 20 were clinically confirmed as positive and 10 as negative, to determine the clinical utility of the BSNFs-chip. In brief, a mixture was prepared with 200 μl NP swab samples, 200 μl of lysis buffer, 50 mg ADH, 10 μl RNase-free DNase, and added PBS to reach a total volume of approximately 650 μl. This mixture was then injected into the BSNFs-chip at a rate of 100 μl per minute using a syringe pump. After pathogen lysis and RNA enrichment by incubating at room temperature for 15 min, an air-filled syringe was utilized to clear out debris of clinical samples and unused ADH from the reaction. Subsequently, any remaining impurities were thoroughly removed using 1 ml of PBS and air. The concentrated RNA derived from the concentrated pathogens were efficiently collected at a flow rate of 25 μl per minute, using 100 μl of pH 10–11 elution buffer. It was confirmed that a higher NA yield was obtained at a flow rate of 25 μl min$^{-1}$ compared to 50 μl min$^{-1}$ on the BSNFs-chip (Supplementary Fig. 11). All the eluted NAs were stored at either −20 or −80 °C for future use. All used microfluidic chips were disposed of after a single use to avoid contamination risks, and the eluted NAs were stored at either −20 or −80 °C for future use. All primers and probes for the qPCR-based methods were synthesized by BIONICS (Seoul, Korea) and are listed in Supplementary Table 2.

## Clinical specimens

The clinical specimens were collected from Asan Medical Center (Seoul, Korea) between March and November 2022. The median age of the COVID-19 positive participants was 61 years (interquartile range 48–66 years) with 45% male, while the COVID-19 negative participants had a median age of 60 years (interquartile range 45–64 years) with 50% male. All patients were enrolled upon confirmation of SARS-CoV-2 infection through nasopharyngeal (NP) RT-PCR. Following enrollment, weekly RT-PCR tests were conducted on respiratory samples, including NP swabs, saliva, or sputum, for up to 12 weeks. If RT-PCR results remained positive after this period, testing continued weekly until two consecutive negative results were obtained. The study protocol was reviewed and approved by the Institutional Review Board of Asan Medical Center (IRB-2022-1054). The research process adhered to the ethical standards for medical research involving human subjects. Participants provided written informed consent before taking part in the study, with the consent form explicitly stating that demographic and clinicopathological information would be used for academic research and potential publication. Participants were not offered any financial compensation for their participation. The study was designed to improve detection sensitivity regardless of patient sex or gender, sex and/or gender were not considered as influencing factors.

In this study, a total of 30 nasopharyngeal (NP) swab samples were used to validate the performance of the BSNFs-chip and PCR-free detection method. These clinical samples comprised of 20 samples from COVID-19 positive patients and 10 samples from patients suspected to have COVID-19, but were later confirmed as negative. All the clinical samples underwent heat inactivation at 60 °C for 30 min and were subsequently stored at −80 °C until they were used. This study

was given ethical approval by the Institutional Review Board of the Asan Medical Center (IRB No. 2022-0297), and all the participants in this study provided informed consent. To isolate the viral RNA, 200 μl of each NP swab sample was used. The isolation was carried out with both the QIAamp Viral RNA Mini Kit (Qiagen) and the BSNFs-chip. In both methods, the viral RNAs were obtained using 100 μl of elution buffer and were then stored at −80 °C until they were further used for analysis or testing.

## Synthesis of the lanthanide-doped nanoparticles (LnNPs)

The core LnNPs were synthesized through thermal decomposition of lanthanide acetate precursors. First, 0.4 mmol of $Y(CH_3CO_2)_3$ (99.9%, Sigma-Aldrich) was mixed with 3 ml oleic acid (90%, Sigma-Aldrich) and 7 ml 1-octadecene (90%, Sigma-Aldrich), and the resulting mixture was heated to 150 °C for 60 min. The solution was then cooled down to 50 °C, and 1 mmol of NaOH (≥98%, Sigma-Aldrich) in methanol and 1.6 mmol of $NH_4F$ (≥99.9%, Sigma-Aldrich) in methanol (≥99.8%, Sigma-Aldrich) were added. The mixture was stirred, and the residual methanol was evaporated by heating the solution to 100 °C and keeping it under vacuum. To heat the solution in an argon environment, the vacuum-argon state was changed three times, and then the argon state was maintained. The resulting core nanoparticles were collected by centrifugation and washed after the resulting solution was heated to 300 °C, held for 1 h, and then cooled to room temperature. The resulting product was re-dispersed in cyclohexane (≥99%, Sigma-Aldrich) for later synthesis. The synthesis of each batch of nanoparticles followed a similar procedure to that used for the core nanoparticles. $Ln(CH_3CO_2)_3$ (Ln = Y, Yb, and Tm, total 0.2 mmol) (99.9%, Sigma-Aldrich) and $Y(CH_3CO_2)_3$ (0.1 mmol) were used for the synthesis of $NaYF_4@NaYF_4$,Yb,Tm and $NaYF_4@NaYF_4$,Yb,Tm@$NaYF_4$, respectively.

## Characterization of the LRET donor

The morphology of the LnNPs was analyzed on a JEM-2100F (JEOL Ltd., Japan) installed at Hanyang LINC3.0 Analytical Equipment Center (Hanyang University, Seoul, Republic of Korea) at an accelerating voltage of 200 kV. The XRD patterns of the LnNPs were characterized by an XRD-7000 diffractometer. The Fourier transform infrared (FT-IR) spectra of the LnNPs were obtained by using a Nicolet iS50 FT-IR spectrophotometer (Thermo Fisher Scientific Co., USA). The hydrodynamic diameter and zeta potential were measured by a Zetasizer Nano ZSP (Malvern Co., UK). The photoluminescence (PL) emission spectra were recorded by a spectrometer (Andor, Kymera 193i) and an intensified sCMOS detector (Andor, ISTAR-SCMOS-18F-73) under external excitation at 980 nm provided by a 980 nm laser diode (Changchun New Industries Optoelectronics Tech. CO., China, MDL-III-980-2W). The PL lifetime was measured using a photomultiplier tube detector (H10721-01; Hamamatsu, Shizuoka, Japan) attached to the spectrometer and a digital oscilloscope (Rhode & Schwarz, Munich, Germany, RTM3002) under excitation at 980 nm using pulsed laser (optical parametric oscillator (OPO) laser, Q-switched Nd-YAG laser, EKSPLA NT342). The PL emission wavelength was selectively measured using a bandpass filter (Semrock, ff-01-800/12-25) and shortpass filter (Semrock, ff-01-950/sp-25) placed in front of the detectors.

## Surface modification of the LnNPs

The LnNPs (15 mg) were dissolved in tetrahydrofuran (≥99.9%, Sigma-Aldrich), and simultaneously, 50 mg of dopamine hydrochloride (≥99.9%, Sigma-Aldrich) was dissolved in water. The solutions were added to the flask and heated to 50 °C with vigorous stirring. After 5 h of incubation under an argon environment, hydrochloric acid (37%, Sigma-Aldrich) solution (1 M) was added, and amine-modified LnNPs were obtained by several washing steps. For preparing maleimide-modified LnNPs, the amine-modified LnNPs (1 mg) and sulfo-SMCC (sulfosuccinimidyl 4-(N-maleimidomethyl) cyclohexane-1-carboxylate,

Thermo Fisher Scientific, Waltham, MA, USA) were dispersed in 10 mM HEPES buffer (pH 7.4) (1 M, Gibco). The resultant solution of maleimide-modified LnNPs was collected through several washing steps after 5 h of incubation.

To prepare DNA oligo-modified LnNPs were obtained using a thiol-maleimide click reaction. Free thiol-modified DNA was prepared using oligo in the disulfide form by the following method. To produce free thiol groups, 20 µl of 100 µM disulfide DNA (Genotech, Daejeon, Korea) was mixed with 20 µl of 5 mM tris(2-carboxyethyl) phosphine hydrochloride (TCEP, Sigma-Aldrich) and the mixture was reacted for 30 min at room temperature. Following this, DNA conjugation reaction was performed overnight at room temperature after adding the thiol-modified DNAs to the maleimide-modified LnNPs. The DNA oligo-modified LnNPs (denoted as LRET donor) were obtained by repeated centrifugation three times and dispersed in 150 µl of HEPES buffer. The S-gene was employed as the target gene sequence for LRET-based detection of SARS-CoV-2 in this study. We employed specific DNA oligonucleotides targeting the part of the SARS-CoV-2 S-gene, and these oligonucleotides had a short length, enabling successful energy transfer. Sequences of the LRET donor and acceptor are listed in Supplementary Table 6.

### LRET-based viral RNA detection

We evaluated analytical sensitivity of the LRET assay using stock SARS-CoV-2 RNA solution isolated by QIAamp Viral RNA Mini Kit (Cat no. 52906, Qiagen). SARS-CoV-2 RNA solution (15 µl) were mixed with the LRET donor (2 µg) and the LRET acceptor (10 pmol, DNA modified IR800 dye) (Integrated DNA Technology, IDT, USA) in HEPES buffer (10 mM, pH 6.2) (total volume = 150 µl) and incubated at room temperature with 600 rpm shaking for 10 min. After the incubation, the PL intensities of the LRET donor in the mixture were measured by the intensified sCMOS detector under external excitation at 980 nm. In the presence of target RNA, the LRET donor and acceptor are brought into close proximity by oligo hybridization between complementary pairs, resulting in quenching of the LRET donor luminescence by the acceptor. From emission spectra, the relative intensity was calculated by the equation below:

$$\text{Relative intensity} = \frac{I_0 - I_x}{I_0} \tag{1}$$

$I_0$ is the PL intensity of the LRET donor and $I_x$ is the PL intensity after incubating with the LRET acceptor in presence of different concentrations of SARS-CoV-2 RNA.

The specificity of the LRET assay was determined using target RNA (SARS-CoV-2) and non-target RNAs including human coronavirus OC43 (hCoV-OC43), hCoV-229E, hCoV-NL63, and H3N2 Influenza A virus (IAV). The hCoV-OC43 and hCoV-229E RNAs were provided by the Korea Bank for Pathogenic Viruses (Seoul, Korea). The hCoV-NL63 RNA was provided by the National Culture Collection for Pathogens (Cheongju, Korea). H3N2 IAV (A/Brisbane/10/2007) RNA was provided by the Korea Research Institute of Bioscience and Biotechnology (KRIBB, Daejeon, Korea).

We validated the clinical applicability of the LRET assay using 30 clinical samples including 20 COVID-19 positive patients and 10 healthy controls. 15 µl of viral RNA samples isolated by the QIAamp Viral RNA Mini Kit (Qiagen) and the BSNFs-chip were mixed with the LRET donor (2 µg) and the LRET acceptor (10 pmol) in HEPES buffer (10 mM, pH 6.2) (total volume = 150 µl). Then, the LRET-based SARS-CoV-2 RNA detection was performed as described above. The cut-off value of relative intensity was determined by applying optimal combinations of clinical sensitivity and specificity from ROC curve based on the Youden index point.

### Statistical analysis

Statistical analyses were performed using Origin Pro 2016 and IBM SPSS statistical package (version 27). The mean and ± standard deviations were calculated for each data point from at least triplicate measurement. LODs and linear ranges were determined using linear regression methods, which included assessing the line slope and the standard deviation of the intercept. The statistical significance of differences between SARS-CoV-2 positive samples and negative samples was assessed using a two-tailed unpaired t-test (*$p < 0.05$, **$p < 0.01$, ***$p < 0.001$, and ****$p < 0.0001$).

### Reporting summary

Further information on research design is available in the Nature Portfolio Reporting Summary linked to this article.

## Data availability

Any additional requests for information can be directed to, and will be fulfilled by, the corresponding authors. Source data are provided with this paper.

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

## Acknowledgements

This research was supported by the National Research Foundation of Korea (NRF) and NST grants funded by the Korean government (MSIT) (NRF-2021M3E5E3080381, NRF-2023R1A2C2003128, CRC-20-0-KIST, J.L.) and KRIBB Research Initiative Program (KGM5472413, T.K.). This

work was also supported in part of by Brain Korea 21 (BK21) FOUR program (M.L. and Y.S.).

## Author contributions

E.J. performed the synthesis of the biporous nanostructures, the overall experiments, and the data analysis. B.K. and M.L. performed the enrichment/isolation experiments and the data analysis. S.K. performed the LRET assay and the data analysis. E.J., B.K., and S.K. wrote the original manuscript. E.J. performed the simulations and J.K. and R.K. performed the data analysis. Y.Y. performed the particle proximity test and the data analysis. H.J., S.-H.K., T.K., and S.K.K. provided the clinical samples and the data analysis. J.L. and Y.S. conceived and designed the study. R.K., Y.S., and J.L. are responsible for supervision of project administration, funding acquisition, and writing-review & editing.

## Competing interests

The authors declare no competing interests.
