## [Peer Review File · Nature Communications]

REVIEWER COMMENTS

Reviewer #1 (Remarks to the Author):

The authors developed a new approach for molecule separation based on porous structures. They designed and tested a porous structure with two levels of porosity and integrated the new material into a microfluidic system. Flow analysis was performed to compare the new structures with a flat surface. It was found that the porous structure increases the mixing of the fluid, improving the slip conditions of the flow and the “molecule capture” ability. This property was claimed as the one responsible for the superior performance of the sensor when compared to standard techniques. The manuscript is well written, and the figures are representative of the discussion. This reviewer recommends publication after revising some important points that it is believed to be the main claims of this work, and responsible for the superior capturing performance of the sensor.

1. “The two BSNFs (top and bottom layers) account for a total height of approximately 800 nm within the microchannel. Notably, the BSNF represents a thin coating, occupying only 0.27% of the total microchannel height of 300 μm ”.

a. Was the designed optimized?

b. Please discuss how the dimensions were selected or how they influence on the sensor performance, in particular related to the flow analysis.

c. Currently the features occupy 0.27% of the volume, does this influence on the overall performance? If they would occupy more volume, it would be more likely to improve mixing and molecule separation.

2. Please discuss how the porosity (area) influences in the overall performance. In figure 2 a detail numerical analysis of the surface area is demonstrated, what is the practical effects of these numbers?

a. Were different porosities tested? It is likely that the porosity would influence on the sensor ability to capture the target molecules.

b. Please discuss if it is possible to control difference ratios and density of porosity in the proposed fabrication protocol and how the different structures would influence the sensor flow and performance.

3. “However, the BSNF, with its enhanced slip flow, can decrease this distance to 22.5 μm (approximately 61% reduction), indicating that the targets can approach and be captured by the BSNF more easily.”

a. Regarding the particle proximity test, please discuss the reason of using fluorescent particles with 10um diameter. It seems difficult to infer any effect of the surface (300 nm) on the particle flow due to the large difference of scale.

b. Please discuss compared to the nanostructures size, the decrease in distance to 22.5 um is not much significant. It is as far as the initial condition without the features in the system. The scales of particles and features should be similar so that a meaningful decreasing in distance can be observed.

Reviewer #2 (Remarks to the Author):

In this paper the authors developed a biporous silica nanofilms chip for pathogen and nucleic acid enrichment and isolation. They coupled their BNSFs-chip with the luminescence resonance energy transfer (LRET) assay for PCR-free analysis of COVID-19 samples and demonstrated that their method achieved a 100-fold lower LOD. The paper can be considered for publication in Nature Communications after revisions based on comments bellow.

- 1) It is not clear how the chip isolate SARS-Cov-2 virus specifically? Amine-modified BNSF chip can also capture any other viruses or vesicles electrostatically. They need to clearly explain and demonstrate specificity using non target similar viruses.
- 2) There is a similar concern about for nucleic acids. Any nucleic acids can also be trapped electrostatically at the amin-modified chips. Specificity of the chip is in question.
- 3) What is the total time required from sample collection to get the answer?
- 4) It is good that their chip can be reused. Is there any change in performance after reuse? How many times a chip can be reused?
- 5) How long the chip is stable? How stable the integrated chip when coupled with LRET assay?
- 6) Their chip could analyze 10-fold diluted samples. What is performance of the chip if they work with undiluted sample. Any device that can analyze undiluted samples are preferable.

Reviewer #3 (Remarks to the Author):

This manuscript was entitled with "Biporous silica nanostructure-induced nanovortex in a microfluidic channel for nucleic acid enrichment, isolation, and PCR-free detection". It introduces a biporous silica nanofilms-embedded sample preparation chip (BSNFs-chip) for pathogen and NA

enrichment/isolation. This design of BSNFs-chip improves the performance through enhancing the surface area and promote the formation of nanovortex, and showing good performance in the clinical results. Therefore, I recommend this work publication after undergoing a major revision focusing on issues like:

1. This design of nanostructures plays a critical role in NA analysis, comprising large and small pore layers. Why design two kinds of pore structures?
2. In this work, the two BSNFs had a total height of approximately 800 nm within the microchannel, and occupied only 0.27% of the total microchannel height of 300 μm . Why chose this height? and this design gives most of sample no chance to contact with 3-aminopropyl(diethoxy) methylsilane. Please compare the effects of different BSNFs heights on sample preparation.
3. The numerical simulation of fluid should no reduce scale (1/100), because it masks the real situation, amplifies the effect of the BSNFs.
4. The sample preparation chips effectively concentrated each cell and virus sample from 1 ml to a final volume of 100 μl . However, the detection limitation had 10-fold lower LOD. It means that this chip only played a role in concentration, thus it should compare different concentration (for example magnetic concentration in the microfluidic) method. and this result demonstrated that the pore structures had a little influence on detection performance.
5. For LRET-based viral RNA detection, the liner detection range should be provided.

REVIEWER COMMENTS

We thank the reviewers for their thoughtful review of the manuscript. We have carefully considered their comments when preparing our revision, which greatly improved the quality of the manuscript. The following text shows our responses to the reviewers' comments.

Reviewer #1 (Remarks to the Author):

The authors developed a new approach for molecule separation based on porous structures. They designed and tested a porous structure with two levels of porosity and integrated the new material into a microfluidic system. Flow analysis was performed to compare the new structures with a flat surface. It was found that the porous structure increases the mixing of the fluid, improving the slip conditions of the flow and the "molecule capture" ability. This property was claimed as the one responsible for the superior performance of the sensor when compared to standard techniques. The manuscript is well written, and the figures are representative of the discussion. This reviewer recommends publication after revising some important points that it is believed to be the main claims of this work, and responsible for the superior capturing performance of the sensor.

- We thank the reviewer for reviewing our manuscript and providing detailed suggestions that helped us improve the quality of the manuscript.

1. "The two BSNFs (top and bottom layers) account for a total height of approximately 800 nm within the microchannel. Notably, the BSNF represents a thin coating, occupying only 0.27% of the total microchannel height of 300 μm ".

a. Was the designed optimized?

Answer. We thank the reviewer for the comment regarding the optimization of BSNF. We conducted experimental optimization in the nanostructure synthesis by varying parameters such as pore size and reaction time, determining the optimal film height for uniform film formation. Additionally, we approached optimization by comparing flow simulations within the nanostructures.

b. Please discuss how the dimensions were selected or how they influence on the sensor performance, in particular related to the flow analysis.

Answer. We thank the reviewer for the comment regarding the influence of dimension selection on sensor performance. In response to the reviewer's comment, we investigated the growth of nanostructures and compared simulations using different heights, as detailed in the following sections.

We observed the growth of nanostructures over reaction time, layer by layer, from small pores in the first layer to large pores in the second layer. These nanostructures displayed gradual growth over the reaction time. However, the small pores in the first layer stopped growing after 3 hours, reaching a maximum height of about 145 nm. Subsequently, they began filling from the bottom, resulting in a reduction of the effective pore channel height. Similarly, the large pores in the second layer maintained their maximum height of 400 nm after 2 hours without further growth. Consequently, we verified that the maximum achievable height for the biporous silica nanofilm was 400 nm (800 nm within the microchannels). The detailed information, including experimental results demonstrating the growth of BSNF, has been added in **Supplementary Fig. 6**, as described in the revised manuscript.

An increase in film height has been anticipated to potentially enhance surface area, offering increased opportunities for molecular binding. However, additional simulations (**Additional Resource. 1**) revealed that increased height did not proportionally correlate with an improvement in chip performance. Specifically, applying a flow rate of 100 $\mu\text{l}/\text{min}$ to simple square-shaped pore structures with heights of 500, 1000, and 1500 nm, demonstrated a

limitation in dynamic flow depth as the height increased, indicating inefficiency. Therefore, based on this study, we conclusively determined the optimal nanostructure height at 400 nm, capped at 800 nm within the microchannel, considering its direct implications on sensor performance in fluid analysis. **Additional resource. 1** related to this result has been added below.

b-1. Please consider the case where the structure height is the same (800 nm total) and the microchannel height is decreased. This is different than only increasing high on the structures. Please discuss.

Answer. Following the Reviewer's additional comment, we conducted simulations at different microchannel heights (i.e., 30 μm and 3 μm). Both the heights showed similar interstitial velocity values (**Additional Resource. 2**). Therefore, in the microchannel height scale of our sample preparation kit, the channel height does not seem to have a significant effect on the nucleic acid enrichment and isolation, unlike the nanostructure height.

We appreciate the reviewer's comment regarding the scale reduction in numerical simulations. As mentioned in the Supplementary Fig.1 and Supplementary Table 1, we compared the effects of the BSNF at a reduced scale of 1/100. Despite intending to conduct simulations at the actual scale as suggested by the reviewer, hardware constraints on our computer posed challenges. As an alternative, we carried out calculations at a reduced 1/10 scale, which proved to be computationally feasible. Interestingly, this indicates that the 1/100 scale doesn't depict the impact of BSNF differently. However, explaining simulations at the 1/10 reduced scale presented visualization challenges within our computer specifications. Therefore, considering these outcomes, we opted for the 1/100 reduced scale to represent visually the effects of BSNF without exaggeration. Additional resources related to this result have been added below.

[Revised main manuscript, page 7] "The cross-view SEM image of the BSNF reveals a total height of approximately 400 nm, with a distinct boundary between the first (height approximately 145 nm) and second layers (height approximately 255 nm) (Fig. 2c and Supplementary Fig. 6)."

[Revised Supplementary Fig. 6]

Supplementary Fig. 6. Cross-sectional SEM images of **a**, PSNF and **b**, BSNF over reaction time.

These nanostructures displayed gradual growth over the reaction time. However, the small pores in the first layer ceased growth after 3 hours, reaching a maximum height of about 145 nm. Subsequently, they began filling from the bottom, resulting in a reduction of the effective pore channel height. Similarly, the large pores in the second layer maintained their maximum height of 400 nm after 2 hours without further growth.

[Additional Resource. 1]

Additional Resource. 1. 3D images of flow velocity field in a simple square-shaped pore structures with heights of **a**, 500 nm, **b**, 1000 nm, and **c**, 1500 nm at a flow rate of 100 $\mu\text{l}/\text{min}$.

[Additional Resource. 2]

Additional Resource. 2. A graph comparing layered interstitial velocity at different microchannel heights.

c. Currently the features occupy 0.27% of the volume, does this influence on the overall performance? If they would occupy more volume, it would be more likely to improve mixing and molecule separation.

Answer. We thank the reviewer for the comment regarding the influence of the feature’s volume on performance. Increasing the volume occupied by the features can indeed enhance mixing and molecule separation by increasing the surface area. However, it is important to note that an increase in height within microchannels can impact fluid flow dynamics. As mentioned in the **Additional Resource. 1**, an increase in height did not proportionally enhance performance in fluid flow. At certain flow rate (100 $\mu\text{l}/\text{min}$), the utilization of the heightened surface area would be limited due to constrained dynamic flow depth. Moreover, as described in our manuscript, despite occupying only 0.27% of the microchannel, our nanostructures improved enrichment and isolation efficiency according to PCR tests. This suggested that even a small volume of nanostructures can positively influence this performance.

Considering that the correlation between height increase, surface area, and performance enhancement has not shown a direct proportion, it is crucial to consider these aspects. Further experiments and modeling to explore the effects of increasing the volume or height of nanostructures on performance enhancement will be valuable and will provide deeper insight into the interrelationships.

2. Please discuss how the porosity (area) influences in the overall performance. In figure 2 a detail numerical analysis of the surface area is demonstrated, what is the practical effects of these numbers?

a. Were different porosities tested? It is likely that the porosity would influence on the sensor ability to capture the target molecules.

b. Please discuss if it is possible to control difference ratios and density of porosity in the proposed fabrication protocol and how the different structures would influence the sensor flow and performance.

Answer. We thank the reviewer for the comments regarding the control of porosity (area) in fabrication protocol and the sensor performance. Accordingly, we explored the nanostructure porosity by changing the ratio of the surfactants, as demonstrated in **Additional Resource. 3**. The following sections have explained the results in comprehensive details.

The porosity of the nanostructures can be controlled by altering the micelle template sizes which is controlled by changing the ratio of the surfactants with opposite charges. In the case of small pores, the micelle template possesses a more positive surface charge, facilitating better interaction with the O₂ plasma-treated negatively charged surface, which results in a well-formed initial layer. However, synthesis of films with larger pore sizes greater than 60 nm presented challenges, as the surface charge of the micelle template neutralized due to increased intrusion of anionic salicylate ions into the cationic CTAC micelles, subsequently weakening its interaction with the substrate surface. However, we found that leveraging a first layer formed by small pores acted as a seed layer during sol-gel process, facilitating the formation of large pores in the subsequent layer—an approach we employed in our design. Therefore, to compare the effect of porosity (surface) on the sensor performance, we synthesized the film with small pores and the film with both small pores and large pores.

In our study, the increased surface area and the generation of nanovortex resulting from the chip's internal porous nanostructures significantly impacted the preparation and enrichment of nucleic acids. This purified and concentrated nucleic acids improve the performance of the sensor (i.e., detection of limit and reliability). To evaluate the influence of surface area, we generated virtual models similar to SEM data, calculating and comparing surface areas within a 500 nm x 500 nm space. According to our calculations, the surface area increase due to small pores was measured at 1244% compared to a flat surface, whereas the increase due to large pores was approximately 619% (1863% - 1244% = 619%), roughly half that of small pores. This indicated that smaller pores enhance the available binding sites for nucleic acids within the same area. However, examination of fluid flow in Figure 3 revealed that nanovortex strength is higher in large pores compared to small pores. This suggests a higher chance for nucleic acids to encounter internal pore surfaces in large pores. The increased surface area implies a greater number of binding sites available for nucleic acids, while the formation of nanovortex provides increased opportunities for binding. Based on these observations, our assessment of the BSNFs-chip's effectiveness in enriching pathogens and NAs using PCR-based methods (Figure 4) was grounded in the high binding probability validated through simulations and experiments.

[Additional Resource. 3]

Additional Resource. 3. SEM images of various nanostructures, the first layer showing different porosity by changing the molar ratio of anionic salicylate ions (Sal-) to cationic cetyltrimethylammonium ions (CTA+).

3. "However, the BSNF, with its enhanced slip flow, can decrease this distance to 22.5 μm (approximately 61% reduction), indicating that the targets can approach and be captured by the BSNF more easily."

a. Regarding the particle proximity test, please discuss the reason of using fluorescent particles with 10 μm diameter. It seems difficult to infer any effect of the surface (300 nm) on the particle flow due to the large difference of scale.

b. Please discuss compared to the nanostructures size, the decrease in distance to 22.5 μm is not much significant. It is as far as the initial condition without the features in the system. The scales of particles and features should be similar so that a meaningful decreasing in distance can be observed.

Answer. We thank the reviewer for the comment regarding the particle proximity test. The followings have explained the results in comprehensive details. First, as the reviewer pointed out, 10 μm diameter particles cannot trace fluid flows in or right above the nanoscale porous structures. However, they can draw fluid streamlines and the boundary layer near the surface qualitatively, which will be changed according to the slip length and/or permeability of the surface. Indeed, the bulk flow profile were totally shifted as the slip length is increased by just adding a thin nanoporous layer on the surface (Fig. 3 in the manuscript). Therefore, as the previous works did (ref. 24 in the manuscript), measuring how much the 10 μm particle can go closer to the surface allows us to identify the degree of slip velocity, permeability of the porous surface, and corresponding thin boundary layer which will bring particles (targets) to the surface more easily. Second, we fully agreed that the smaller trace particles may show us more clear vision of the effects of the nanoporous structures. However, tracking nanoparticles require more specialized method, because a conventional fluorescent microscope (as we used in this experiment) has a limited spatial resolution to detect them¹. Given your comment, we clarified and added these discussions and limits of the particle proximity test in the revised manuscript.

[Revised main manuscript, page 9]

"When we propel the 10 μm fluorescent particles in the solution toward a vertically aligned solid surface under perfect slip conditions with an ideally inviscid flow, the streamline (or particle trajectories) shows a two-dimensional stagnation flow with a single stagnation point at the center of the solid surface. While microscale particles cannot trace fluid flows in or right above the nanoscale porous structures, they can draw fluid streamlines and the boundary layer near the surface qualitatively, which will be changed according to the slip length and/or permeability of the surface."

[Reference]

1. Kazoe, Y., Shibata, K. & Kitamori, T. Super-Resolution Defocusing Nanoparticle Image Velocimetry Utilizing Spherical Aberration for Nanochannel Flows. *Anal. Chem.* **93**, 13260-13267 (2021).

Reviewer #2

In this paper the authors developed a biporous silica nanofilms chip for pathogen and nucleic acid enrichment and isolation. They coupled their BSNFs-chip with the luminescence resonance energy transfer (LRET) assay for PCR-free analysis of COVID19 samples and demonstrated that their method achieved a 100-fold lower LOD. The paper can be considered for publication in Nature Communications after revisions based on comments below.

- We thank the reviewer for reviewing our manuscript and providing detailed suggestions that helped us improve the quality of the manuscript.

1) It is not clear how the chip isolate SARS-Cov-2 virus specifically? Amine-modified BSNF chip can also capture any other viruses or vesicles electrostatically. They need to clearly explain and demonstrate specificity using non target similar viruses.

Answer. We thank the reviewer for the comment regarding the specificity of our BSNFs-chip in isolating SARS-CoV-2. As the reviewer correctly noted, the amine-modified BSNFs-chip is designed to electrostatically capture a range of entities with a negative surface charge, including cells, bacteria, viruses, and vesicles. This capability is exemplified in Fig. 4 of our manuscript, where the HCT116 cell line and SARS-CoV-2 virus are used. The broad-spectrum capture capability of the BSNFs-chip is a significant feature, enabling applications for the enrichment of target biomaterials in biological sample processing. **Supplementary Fig. 10** has been revised to clearly explain this broad-spectrum capture capability.

However, the specific detection of SARS-CoV-2 in our system is achieved through the integration of the PCR-free LRET assay, which plays a role in conferring specificity to our detection system. This assay is designed to target the S gene of SARS-CoV-2, thereby ensuring accurate and specific detection. To address the reviewer's concern about specificity, we conducted additional experiments with viruses similar to SARS-CoV-2, but not targeted by our system (**Supplementary Fig. 16a,b**). These results were instrumental in showing that the LRET assay can distinctly identify SARS-CoV-2 RNA, effectively differentiating it from other viral RNAs. This specificity is crucial for the accurate detection of SARS-CoV-2, ensuring that our system is not confounded by the presence of other viruses.

Furthermore, we recognize the potential of the BSNFs-chip in enriching and isolating vesicles which also carry a negative surface charge. We plan to explore this application in future research, expanding the utility of the BSNFs-chip beyond its current focus on SARS-CoV-2 detection. This detailed information, including the experimental details and results demonstrating the specificity of the LRET assay, has been added to the revised manuscript.

[Revised main manuscript, page 14]

"Future investigations will focus on extending the applicability of these nanomaterials through exploring the micelle aggregation mechanism and precise pore control, including the design of customized nanostructures, while simultaneously aiming to enhance the integration of the BSNFs-chip with the LRET assay to expand their combined utility beyond SARS-CoV-2 detection to a broader range of pathogens. We anticipate that this approach will play an impactful role in advancing the field of rapid and sensitive diagnostic methods for infectious diseases in the years to come."

[Revised main manuscript, page 12]

"Furthermore, the DNA oligos on the surface of the LRET donor and acceptor could hybridize specifically to SARS-CoV-2 RNA with no obvious cross-reactivity to other non-target NAs (Supplementary Fig. 14a). We validated the specificity of the LRET assay using 1 pM of non-target sequences of other common contagious respiratory viruses. The LRET assay successfully distinguished SARS-CoV-2 from human coronavirus OC43 (hCoV-OC43), hCoV-NL63, hCoV-229E, and influenza A virus (IAV) (Supplementary Fig. 14b)."

[Revised main manuscript, page 22]

“The specificity of the LRET assay was determined using target RNA (SARS-CoV-2) and non-target RNAs including human coronavirus OC43 (hCoV-OC43), human coronavirus 229E (hCoV-229E), human coronavirus NL63 (hCoV-NL63), and H3N2 Influenza A virus (IAV). The hCoV-OC43 and hCoV-229E RNAs were provided by the Korea Bank for Pathogenic Viruses (Seoul, Korea). The hCoV-NL63 RNA was provided by the National Culture Collection for Pathogens (Cheongju, Korea). H3N2 IAV (A/Brisbane/10/2007) RNA was provided by the Korea Research Institute of Bioscience and Biotechnology (KRIBB, Daejeon, Korea).”

[Revised Supplementary Fig. 10]

Supplementary Fig. 10. Schematic overviews of the sample preparation process of chips with **a**, BSNFs-chip and **b**, Flat-chip. The amine-modified BSNFs and flat surfaces are designed to electrostatically capture a range of entities with a negative surface charge, including cells, bacteria, viruses, and vesicles.

[Revised Supplementary Fig. 16]

Supplementary Fig. 16. LRET-based detection of SARS-CoV-2 RNA. **a**, Schematic illustration of the LRET-based detection specific to SARS-CoV-2 RNA. **b**, The LRET assay demonstrated specificity for SARS-CoV-2, with no cross-reactivity towards other common respiratory viruses, including human coronavirus OC43 (hCoV-OC43), hCoV-NL63, hCoV-229E, and influenza A virus (IAV). **c**, Emission spectra of the LRET donor with various concentrations of SARS-CoV-2 RNA. **d**, Calibration curve of quenching efficiencies with increasing concentrations of SARS-CoV-2 RNA. A linear fit to the data over this concentration range yielded the relation $(I_0 - I_x)/I_0 = 0.065 \times \log C + 0.094$, where C is the concentration of RNA, with a correlation coefficient $R^2 = 0.998$. All reported values represent the mean \pm SD, $n = 3$. The cut-off value was determined by applying optimal combinations of clinical sensitivity and specificity from ROC curve based on the Youden index point. **e**, Luminescence lifetime curves at 800 nm of LRET donor in the absence or presence of target RNA (under 980 nm excitation).

2) There is a similar concern about for nucleic acids. Any nucleic acids can also be trapped electrostatically at the amin-modified chips. Specificity of the chip is in question.

Answer. We thank the reviewer for the comment regarding the specificity of our BSNFs-chip in trapping nucleic acids (NAs) from SARS-CoV-2. Building on our previous response, the amine-modified BSNFs-chip demonstrates a robust capability for capturing various NAs through electrostatic and covalent binding. This capability is evidenced in our study, particularly in Fig. 4, where we successfully captured both genomic DNA and RNA from the HCT116 cell line, as well as viral RNA from SARS-CoV-2. Notably, the specificity of our system in detecting SARS-CoV-2 hinges on the PCR-free LRET assay. This assay is designed to target the S gene of SARS-CoV-2, thereby ensuring accurate and specific detection. In the revised manuscript, we have included experimental data and results that underscore the LRET assay's effectiveness in identifying SARS-CoV-2 RNA, distinguishing it from other viral RNAs. These details, which are elaborated upon in our response to the first comment, have been explained into the revised manuscript to provide a comprehensive understanding of our system's specificity in detecting SARS-CoV-2.

3) What is the total time required from sample collection to get the answer?

Answer. We thank the reviewer for the comment regarding the total time required for our process, from sample collection to get the answer. Our study demonstrates that the use of the BSNFs-chip for pathogen and NA enrichment/isolation takes approximately 40 minutes. For the detection

of SARS-CoV-2 in clinical samples, our methodology incorporates a sequential procedure. First, we employ the BSNFs-chip for pathogen and NA enrichment/isolation, which takes around 40 minutes using a 200 μ L sample volume. Following this, the PCR-free LRET assay is conducted, requiring an additional 10 minutes. Therefore, the total time from sample collection to result acquisition for SARS-CoV-2 detection is approximately 50 minutes. This streamlined process, combining the BSNFs-chip with the PCR-free LRET assay, not only enhances the efficiency of pathogen detection but also significantly reduces the overall time required for analysis compared to traditional methods. This comprehensive timing information has been detailed in the revised manuscript.

[Revised main manuscript, page 11]

"This strategy begins with the enrichment/isolation of pathogens and NAs using the BSNFs-chip (approximately 40 min), which is followed by the identification of enriched target RNAs using the LRET assay (approximately 10 min), and the entire process from sample collection to obtaining results for SARS-CoV-2 detection is completed within 50 minutes (Fig. 5a)."

4) It is good that their chip can be reused. Is there any change in performance after reuse? How many times a chip can be reused?

Answer. We thank the reviewer for the comment regarding the reusability of our chip. The reusability of the BSNFs-chip represents a significant advancement in terms of both cost-effectiveness and environmental sustainability. In the manuscript, Supplementary Fig. 13 provides a comprehensive analysis of the BSNFs-chip's reusability. Figure S13a presents SEM images demonstrating the structural stability of the BSNF after the sample preparation process, showing that the BSNF maintains its integrity even after sample processing. Figure S13b further analyzes the structural robustness and reusability of the chip in pathogen and NA enrichment/isolation. The Ct values obtained were 27.78 ± 0.13 for the first use, 27.3 ± 0.35 for the second use, and 27.03 ± 0.30 for the third use, indicating high reproducibility. As depicted in these figures, our findings suggest that the BSNFs-chip could potentially be used more than three times, given its robustness and consistent performance. However, we chose to use the BSNFs chip only once to completely avoid contamination risks. This decision was made despite the potential for extended reuse, as our priority was to ensure the highest standards of accuracy in every application. While the chip exhibits excellent structural stability and efficiency, we prioritized contamination control over the possibility of reuse. This approach and the associated data have been added to the revised manuscript.

[Revised main manuscript, page 11]

"The structural stability of the BSNFs-chip was also confirmed even after being used for pathogen and NA enrichment/isolation (Supplementary Fig. 13). This robustness makes the BSNFs-chip suitable for both disposable and reusable applications. Notably, the chip demonstrated consistent performance across multiple uses, with Ct values of 27.78 ± 0.13 for the first use, 27.3 ± 0.35 for the second use, and 27.03 ± 0.30 for the third use, indicating high reproducibility. These results further establish the BSNFs-chip as a promising sample preparation platform for the accurate diagnosis of infectious diseases, highlighting its potential as a reliable tool in the sensitive and precise diagnosis of infectious diseases."

[Revised main manuscript, page 20]

"All used microfluidic chips were disposed of after a single use to avoid contamination risks, and the eluted NAs were stored at either -20 or -80 $^{\circ}$ C for future use."

5) How long the chip is stable? How stable the integrated chip when coupled with LRET assay?

Answer. We thank the reviewer for the comment regarding the stability of BSNFs-chip. In response to the reviewer's comment, we fabricated a new BSNFs-chip and assessed its stability over a two-week period. The stability of BSNF chips can offer a significant advantage in terms of convenience in storage and ensuring reliability and consistency of experimental results. The BSNFs-chips have been utilized under prolonged storage at room temperature post-fabrication. These conditions are consistent across experiments, notably in sample preparation for PCR test and LRET assay. We conducted SEM imaging and FT-IR spectroscopy over a two-week period immediately following chip fabrication, confirming structural integrity and the consistent presence of amine peak around 3350 cm^{-1} . The detailed information, including experimental results demonstrating the stability of BSNFs-chip, has been added in **Supplementary Fig. 12**, as described in the revised manuscript.

[Revised main manuscript, page 11]

“The stability of the BSNFs-chip was confirmed over a two-week period immediately following chip fabrication through SEM imaging and FT-IR spectroscopy (Supplementary Fig. 12).”

[Revised Supplementary Fig. 12]

Supplementary Fig. 12. SEM images and FT-IR spectrums showing the stability of the BSNFs-chip over a two-week period immediately following chip fabrication.

The BSNFs-chip was utilized underwent prolonged storage at room temperature post-fabrication. These conditions were consistent across experiments, notably in sample preparations for PCR test and LRET assay.

6) Their chip could analyze 10-fold diluted samples. What is performance of the chip if they work with undiluted sample. Any device that can analyze undiluted samples are preferable.

Answer. We thank the reviewer for the comment regarding the performance of our BSNFs-chip with undiluted samples. We apologize for any confusion caused by a wording error in the Methods section, specifically, “10-fold dilutions of SARS-CoV-2 RNA solution (15 μ l) were mixed with ...”. We performed analyses using clinical samples in their native, undiluted state, as detailed in Fig. 5 and Supplementary Figs. 17 and 18. Notably, Supplementary Fig. 17 illustrates the qRT-PCR detection of SARS-CoV-2 RNA from nasopharyngeal swab samples, confirming the ability of the chip to process and detect the virus directly from clinical specimens. Additionally, 15 μ l of these samples were mixed with LRET donor, acceptor, and reaction buffer, constituting 10% of the total reaction solution. However, it is noted that the samples were not further diluted to assess the performance of the BSNF chip integrated with the LRET assay. The comparisons in Supplementary Figs. 18a and 18b between our BSNFs-chip and traditional extraction methods, particularly using the LRET assay, further underscore the chip's enhanced sensitivity and reliability with undiluted samples. These findings collectively establish the BSNFs-chip's suitability for practical diagnostic use in clinical settings, confirming its adaptability and effectiveness in handling various sample conditions, including undiluted samples. To help clear understanding, the phrase “10-fold dilutions of” was deleted and a detailed method for LRET-based detection using clinical samples was described in the revised manuscript.

[Revised main manuscript, page 21]

~~10-fold dilutions of~~ SARS-CoV-2 RNA solution (15 μ l), which was isolated by the BSNFs-chip or QIAamp Viral RNA Mini Kit (Cat no. 52906, Qiagen) were mixed with the LRET donor (2 μ g) and the LRET acceptor (10 pmol, DNA modified IR800 dye) (Integrated DNA Technology, IDT) in HEPES buffer (10 mM, pH 6.2) and incubated at room temperature with 600 rpm shaking for 10 minutes.

[Revised main manuscript, page 21]

We validated the clinical applicability of the LRET assay using 30 clinical samples including 20 COVID-19 positive patients and 10 healthy controls. 15 μ l of viral RNA samples isolated by the QIAamp Viral RNA Mini Kit (Qiagen) and the BSNFs-chip were mixed with the LRET donor (2 μ g) and the LRET acceptor (10 pmol) in HEPES buffer (10 mM, pH 6.2) (total volume = 150 μ l). Then, the LRET-based SARS-CoV-2 RNA detection was performed as described above.

Reviewer #3

This manuscript was entitled with "Biporous silica nanostructure-induced nanovortex in a microfluidic channel for nucleic acid enrichment, isolation, and PCR-free detection". It introduces a biporous silica nanofilms-embedded sample preparation chip (BSNFs-chip) for pathogen and NA enrichment/isolation. This design of BSNFs-chip improves the performance through enhancing the surface area and promote the formation of nanovortex, and showing good performance in the clinical results. Therefore, I recommend this work publication after undergoing a major revision focusing on issues like:

- We thank the reviewer for reviewing our manuscript and providing detailed suggestions that helped us improve the quality of the manuscript.

1. This design of nanostructures plays a critical role in NA analysis, comprising large and small pore layers. Why design two kinds of pore structures?

Answer. We thank the reviewer for the comments regarding the control of porosity (area) in fabrication protocol and the sensor performance. Accordingly, we explored the nanostructure porosity by changing the ratio of the surfactants, as demonstrated in **Additional Resource. 1**. The following sections have explained the results in comprehensive details.

The porosity of the nanostructures can be controlled by altering the micelle template sizes which is controlled by changing the ratio of the surfactants with opposite charges. In the case of small pores, the micelle template possesses a more positive surface charge, facilitating better interaction with the O₂ plasma-treated negatively charged surface, which results in a well-formed initial layer. However, synthesis of films with larger pore sizes greater than 60 nm presented challenges, as the surface charge of the micelle template neutralized due to increased intrusion of anionic salicylate ions into the cationic CTAC micelles, subsequently weakening its interaction with the substrate surface. However, we found that leveraging a first layer formed by small pores acted as a seed layer during sol-gel process, facilitating the formation of large pores in the subsequent layer—an approach we employed in our design. Therefore, to compare the effect of porosity (surface) on the sensor performance, we synthesized the film with small pores and the film with both small pores and large pores.

In our study, the increased surface area and the generation of nanovortex resulting from the chip's internal porous nanostructures significantly impacted the preparation and enrichment of nucleic acids. This purified and concentrated nucleic acids improve the performance of the sensor (i.e., detection of limit and reliability). To evaluate the influence of surface area, we generated virtual models similar to SEM data, calculating and comparing surface areas within a 500 nm x 500 nm space. According to our calculations, the surface area increase due to small pores was measured at 1244% compared to a flat surface, whereas the increase due to large pores was approximately 619% (1863% - 1244% = 619%), roughly half that of small pores. This indicated that smaller pores enhance the available binding sites for nucleic acids within the same area. However, examination of fluid flow in Figure 3 revealed that nanovortex strength is higher in large pores compared to small pores. This suggests a higher chance for nucleic acids to encounter internal pore surfaces in large pores. The increased surface area implies a greater number of binding sites available for nucleic acids, while the formation of nanovortex provides increased opportunities for binding. Based on these observations, our assessment of the BSNFs-chip's effectiveness in enriching pathogens and NAs using PCR-based methods (Figure 4) was grounded in the high binding probability validated through simulations and experiments.

[Additional Resource. 1]

Additional Resource. 1. SEM images of various nanostructures, the first layer showing different porosity by changing the molar ratio of anionic salicylate ions (Sal⁻) to cationic cetyltrimethylammonium ions (CTA⁺).

2. In this work, the two BSNFs had a total height of approximately 800 nm within the microchannel, and occupied only 0.27% of the total microchannel height of 300 μm. Why chose this height? and this design gives most of sample no chance to contact with 3aminopropyl(diethoxy) methylsilane. Please compare the effects of different BSNFs heights on sample preparation.

Answer. We thank the reviewer for the comment regarding the influence of dimension selection on sensor performance. In response to the reviewer's comment, we investigated the growth of nanostructures and compared simulations using different heights, as detailed in the following sections.

We observed the growth of nanostructures over reaction time, layer by layer, from small pores in the first layer to large pores in the second layer. These nanostructures displayed gradual growth over the reaction time. However, the small pores in the first layer ceased growth after 3 hours, reaching a maximum height of about 145 nm. Subsequently, they began filling from the bottom, resulting in a reduction of the effective pore channel height. Similarly, the large pores in the second layer maintained their maximum height of 400 nm after 2 hours without further growth. Consequently, we verified that the maximum achievable height for the biporous silica nanofilm was 400 nm (800 nm within the microchannels). The detailed information, including experimental results demonstrating the growth of BSNF, has been added in **Supplementary Fig. 6**, as described in the revised manuscript.

An increase in film height has been anticipated to potentially enhance surface area, offering increased opportunities for molecular binding. However, additional simulations (**Additional Resource. 2**) revealed that increased height did not proportionally correlate with an improvement in chip performance. Specifically, applying a flow rate of 100 μl/min to simple square-shaped pore structures with heights of 500, 1000, and 1500 nm, demonstrated a limitation in dynamic flow depth as the height increased, indicating inefficiency. Therefore, based on this study, we conclusively determined the optimal nanostructure height at 400 nm, capped at 800 nm within the microchannel, considering its direct implications on sensor performance in fluid analysis. **Additional resource. 2** related to this result has been added below.

[Revised main manuscript, page 7] "The cross-view SEM image of the BSNF reveals a total height of approximately 400 nm, with a distinct boundary between the first (height approximately 145 nm) and second layers (height approximately 255 nm) (Fig. 2c and Supplementary Fig. 6)."

[Revised Supplementary Fig. 6]

Supplementary Fig. 6. Cross-sectional SEM images of **a**, PSNF and **b**, BSNF over reaction time.

These nanostructures displayed gradual growth over the reaction time. However, the small pores in the first layer ceased growth after 3 hours, reaching a maximum height of about 145 nm. Subsequently, they began filling from the bottom, resulting in a reduction of the effective pore channel height. Similarly, the large pores in the second layer maintained their maximum height of 400 nm after 2 hours without further growth.

[Additional Resource. 2]

Additional Resource. 2. 3D images of flow velocity field in a simple square-shaped pore structures with heights of **a**, 500 nm, **b**, 1000 nm, and **c**, 1500 nm at a flow rate of 100 $\mu\text{l}/\text{min}$.

3. The numerical simulation of fluid should no reduce scale (1/100), because it masks the real situation, amplifies the effect of the BSNFs.

Answer. We appreciate the reviewer's comment regarding the scale reduction in numerical simulations. As mentioned in the Supplementary Fig.1 and Supplementary Table 1, we compared the effects of the BSNF at a reduced scale of 1/100. Despite intending to conduct simulations at the actual scale as suggested by the reviewer, hardware constraints on our computer posed challenges (**Additional Resource. 3**). As an alternative, we carried out calculations at a reduced 1/10 scale, which proved to be computationally feasible. Interestingly, both the 1/10 and 1/100 reduced scales showed similar interstitial velocity values (**Additional Resource. 4**). This indicates that the 1/100 scale doesn't depict the impact of BSNF differently. However, explaining simulations at the 1/10 reduced scale presented visualization challenges within our computer specifications. Therefore, considering these outcomes, we opted for the 1/100 reduced scale to represent visually the effects of BSNF without exaggeration. Additional resources related to this result have been added below.

[Additional Resource. 3]

```
-----  
System Information  
-----  
Time of this report: 11/24/2023, 18:50:46  
Machine name: DESKTOP-LGUVQ7G  
Machine Id: {36A8B246-6DD6-4D53-9656-93B4863990BB}  
Operating System: Windows 11 Pro for Workstations 64-bit (10.0, Build 22621) (22621.ni_release.220506-1250)  
Language: Korean (Regional Setting: Korean)  
System Manufacturer: HP  
System Model: HP Z6 G4 Workstation  
BIOS: P60 v02.91 (type: UEFI)  
Processor: Intel(R) Xeon(R) Silver 4214R CPU @ 2.40GHz (48 CPUs), ~2.4GHz  
Memory: 262144MB RAM  
Available OS Memory: 261848MB RAM  
Page File: 20780MB used, 361993MB available  
Windows Dir: C:\windows  
DirectX Version: DirectX 12  
DX Setup Parameters: Not found  
User DPI Setting: 96 DPI (100 percent)  
System DPI Setting: 96 DPI (100 percent)  
DWM DPI Scaling: Disabled  
Miracast: Available, no HDCP  
Microsoft Graphics Hybrid: Not Supported  
DirectX Database Version: 1.4.7  
DxDiag Version: 10.00.22621.0001 64bit Unicode  
  
-----  
Display Devices  
-----  
Card name: NVIDIA RTX A4000  
Manufacturer: NVIDIA  
Chip type: NVIDIA RTX A4000
```

Additional Resource. 3. Capture image showing system information and display devices used in numerical simulations.

[Additional Resource. 4]

Additional Resource. 4. A graph comparing layered interstitial velocity at scales reduced to 1/10 and 1/100.

4. The sample preparation chips effectively concentrated each cell and virus sample from 1 ml to a final volume of 100 μ l. However, the detection limitation had 10-fold lower LOD. It means that this chip only played a role in concentration, thus it should compare different concentration (for example magnetic concentration in the microfluidic) method. and this result demonstrated that the pore structures had a little influence on detection performance.

Answer. We thank the reviewer for the comment regarding the 10-fold lower LOD of our BSNFs-chip. As illustrated in Fig. 4c of our manuscript, for DNA, both the BSNFs- and PSNFs-chips showed a 10-fold lower LOD (1×10^1 cells/ml) compared to the Flat-chip, and a 100-fold lower LOD

than conventional methods. For RNA, encompassing both genomic RNA and viral RNA, the BSNFs-chip exhibited a 10-fold higher sensitivity (LOD of 1×10^1 cells/ml for genomic RNA and 0.96×10^0 PFU/ml for viral RNA) than the Flat- and PSNFs-chips, and a 100-fold higher sensitivity compared to conventional methods, as detailed in Fig. 4d and 4e. This result is not simply due to using a larger sample volume to achieve a 10-fold improvement in sensitivity. Instead, it underscores the BSNFs-chip's capability to surpass the limitations of detection methods at low concentration levels. This is evidenced by its ability to achieve sensitivity levels 10 to 100 times greater than those of conventional methods, a significant advancement attributable to its unique biporous silica nanostructures and the nanovortex effect. These innovative design features are crucial for detecting pathogens and NAs at substantially lower concentrations than possible with conventional methods.

5. For LRET-based viral RNA detection, the linear detection range should be provided.

Answer. Thanks for the Reviewer's attentive comments. The LRET assay demonstrated a high correlation coefficient (R^2) of 0.99783 ranging from 10^{-1} to 10^3 PFU of SARS-CoV-2 RNA, as shown in Supplementary Fig. 16d. Following the Reviewer's comment, we have added more detail on the linear detection range in the revised manuscript.

Supplementary Fig. 16 c, Emission spectra of the LRET donor with various concentrations of SARS-CoV-2 RNA. **d**, Calibration curve of quenching efficiencies with increasing concentrations of SARS-CoV-2 RNA. A linear fit to the data over this concentration range yielded the relation $(I_0 - I_x)/I_0 = 0.065 \times \log C + 0.094$, where C is the concentration of RNA, with a correlation coefficient $R^2 = 0.998$.

[Revised main manuscript, page 12]

The relative intensities displayed a linear relationship with the logarithmic concentration of SARS-CoV-2 RNA ranging from 10^{-1} to 10^3 PFU, with a high correlation coefficient (R^2) of 0.998 (Fig. 5c and Supplementary Fig. 16c,d).

REVIEWERS' COMMENTS

Reviewer #1 (Remarks to the Author):

The authors addressed all comments from the reviewers.

This reviewer is satisfied with the new information and data added to the manuscript.

From the comments and responses from the authors, this reviewer has one more comment regarding the sensor-device optimization:

1- Additional Resource. 1: please consider the case where the structure height is the same (800 nm total) and the microchannel height is decreased. This is different than only increasing high on the structures. Please discuss.

Reviewer #2 (Remarks to the Author):

The concerns I raised have been effectively addressed by the authors, and they have made improvements to the manuscript. I suggest considering the paper for publication in its current form.

REVIEWER COMMENTS

We thank the reviewers for their thoughtful review of the manuscript. We have carefully considered their comments when preparing our revision, which greatly improved the quality of the manuscript. The following text shows our responses to the reviewers' comments.

Reviewer #1 (Remarks to the Author):

The authors developed a new approach for molecule separation based on porous structures. They designed and tested a porous structure with two levels of porosity and integrated the new material into a microfluidic system. Flow analysis was performed to compare the new structures with a flat surface. It was found that the porous structure increases the mixing of the fluid, improving the slip conditions of the flow and the "molecule capture" ability. This property was claimed as the one responsible for the superior performance of the sensor when compared to standard techniques. The manuscript is well written, and the figures are representative of the discussion. This reviewer recommends publication after revising some important points that it is believed to be the main claims of this work, and responsible for the superior capturing performance of the sensor.

- We thank the reviewer for reviewing our manuscript and providing detailed suggestions that helped us improve the quality of the manuscript.

1. "The two BSNFs (top and bottom layers) account for a total height of approximately 800 nm within the microchannel. Notably, the BSNF represents a thin coating, occupying only 0.27% of the total microchannel height of 300 μm ".

a. Was the designed optimized?

Answer. We thank the reviewer for the comment regarding the optimization of BSNF. We conducted experimental optimization in the nanostructure synthesis by varying parameters such as pore size and reaction time, determining the optimal film height for uniform film formation. Additionally, we approached optimization by comparing flow simulations within the nanostructures.

b. Please discuss how the dimensions were selected or how they influence on the sensor performance, in particular related to the flow analysis.

Answer. We thank the reviewer for the comment regarding the influence of dimension selection on sensor performance. In response to the reviewer's comment, we investigated the growth of nanostructures and compared simulations using different heights, as detailed in the following sections.

We observed the growth of nanostructures over reaction time, layer by layer, from small pores in the first layer to large pores in the second layer. These nanostructures displayed gradual growth over the reaction time. However, the small pores in the first layer stopped growing after 3 hours, reaching a maximum height of about 145 nm. Subsequently, they began filling from the bottom, resulting in a reduction of the effective pore channel height. Similarly, the large pores in the second layer maintained their maximum height of 400 nm after 2 hours without further growth. Consequently, we verified that the maximum achievable height for the biporous silica nanofilm was 400 nm (800 nm within the microchannels). The detailed information, including experimental results demonstrating the growth of BSNF, has been added in **Supplementary Fig. 6**, as described in the revised manuscript.

An increase in film height has been anticipated to potentially enhance surface area, offering increased opportunities for molecular binding. However, additional simulations (**Additional Resource. 1**) revealed that increased height did not proportionally correlate with an improvement in chip performance. Specifically, applying a flow rate of 100 $\mu\text{l}/\text{min}$ to simple square-shaped pore structures with heights of 500, 1000, and 1500 nm, demonstrated a

limitation in dynamic flow depth as the height increased, indicating inefficiency. Therefore, based on this study, we conclusively determined the optimal nanostructure height at 400 nm, capped at 800 nm within the microchannel, considering its direct implications on sensor performance in fluid analysis. **Additional resource. 1** related to this result has been added below.

b-1. Please consider the case where the structure height is the same (800 nm total) and the microchannel height is decreased. This is different than only increasing high on the structures. Please discuss.

Answer. Following the Reviewer's additional comment, we conducted simulations at different microchannel heights (i.e., 30 μm and 3 μm). Both the heights showed similar interstitial velocity values (**Additional Resource. 2**). Therefore, in the microchannel height scale of our sample preparation kit, the channel height does not seem to have a significant effect on the nucleic acid enrichment and isolation, unlike the nanostructure height.

We appreciate the reviewer's comment regarding the scale reduction in numerical simulations. As mentioned in the Supplementary Fig.1 and Supplementary Table 1, we compared the effects of the BSNF at a reduced scale of 1/100. Despite intending to conduct simulations at the actual scale as suggested by the reviewer, hardware constraints on our computer posed challenges (**Additional Resource. 3**). As an alternative, we carried out calculations at a reduced 1/10 scale, which proved to be computationally feasible. Interestingly,. This indicates that the 1/100 scale doesn't depict the impact of BSNF differently. However, explaining simulations at the 1/10 reduced scale presented visualization challenges within our computer specifications. Therefore, considering these outcomes, we opted for the 1/100 reduced scale to represent visually the effects of BSNF without exaggeration. Additional resources related to this result have been added below.

[Revised main manuscript, page 7] "The cross-view SEM image of the BSNF reveals a total height of approximately 400 nm, with a distinct boundary between the first (height approximately 145 nm) and second layers (height approximately 255 nm) (Fig. 2c and Supplementary Fig. 6)."

[Revised Supplementary Fig. 6]

Supplementary Fig. 6. Cross-sectional SEM images of **a**, PSNF and **b**, BSNF over reaction time.

These nanostructures displayed gradual growth over the reaction time. However, the small pores in the first layer ceased growth after 3 hours, reaching a maximum height of about 145 nm. Subsequently, they began filling from the bottom, resulting in a reduction of the effective pore channel height. Similarly, the large pores in the second layer maintained their maximum height of 400 nm after 2 hours without further growth.

[Additional Resource. 1]

Additional Resource. 1. 3D images of flow velocity field in a simple square-shaped pore structures with heights of **a**, 500 nm, **b**, 1000 nm, and **c**, 1500 nm at a flow rate of 100 $\mu\text{l}/\text{min}$.

[Additional Resource. 2]

Additional Resource. 2. A graph comparing layered interstitial velocity at different microchannel heights.

c. Currently the features occupy 0.27% of the volume, does this influence on the overall performance? If they would occupy more volume, it would be more likely to improve mixing and molecule separation.

Answer. We thank the reviewer for the comment regarding the influence of the feature's volume on performance. Increasing the volume occupied by the features can indeed enhance mixing and molecule separation by increasing the surface area. However, it is important to note that an increase in height within microchannels can impact fluid flow dynamics. As mentioned in the **Additional Resource. 1**, an increase in height did not proportionally enhance performance in fluid flow. At certain flow rate (100 $\mu\text{l}/\text{min}$), the utilization of the heightened surface area would be limited due to constrained dynamic flow depth. Moreover, as described in our manuscript, despite occupying only 0.27% of the microchannel, our nanostructures improved enrichment and isolation efficiency according to PCR tests. This suggested that even a small volume of nanostructures can positively influence this performance.

Considering that the correlation between height increase, surface area, and performance enhancement has not shown a direct proportion, it is crucial to consider these aspects. Further experiments and modeling to explore the effects of increasing the volume or height of nanostructures on performance enhancement will be valuable and will provide deeper insight into the interrelationships.

2. Please discuss how the porosity (area) influences in the overall performance. In figure 2 a detail numerical analysis of the surface area is demonstrated, what is the practical effects of these numbers?

a. Were different porosities tested? It is likely that the porosity would influence on the sensor ability to capture the target molecules.

b. Please discuss if it is possible to control difference ratios and density of porosity in the proposed fabrication protocol and how the different structures would influence the sensor flow and performance.

Answer. We thank the reviewer for the comments regarding the control of porosity (area) in fabrication protocol and the sensor performance. Accordingly, we explored the nanostructure porosity by changing the ratio of the surfactants, as demonstrated in **Additional Resource. 3**. The following sections have explained the results in comprehensive details.

The porosity of the nanostructures can be controlled by altering the micelle template sizes which is controlled by changing the ratio of the surfactants with opposite charges. In the case of small pores, the micelle template possesses a more positive surface charge, facilitating better interaction with the O₂ plasma-treated negatively charged surface, which results in a well-formed initial layer. However, synthesis of films with larger pore sizes greater than 60 nm presented challenges, as the surface charge of the micelle template neutralized due to increased intrusion of anionic salicylate ions into the cationic CTAC micelles, subsequently weakening its interaction with the substrate surface. However, we found that leveraging a first layer formed by small pores acted as a seed layer during sol-gel process, facilitating the formation of large pores in the subsequent layer—an approach we employed in our design. Therefore, to compare the effect of porosity (surface) on the sensor performance, we synthesized the film with small pores and the film with both small pores and large pores.

In our study, the increased surface area and the generation of nanovortex resulting from the chip's internal porous nanostructures significantly impacted the preparation and enrichment of nucleic acids. This purified and concentrated nucleic acids improve the performance of the sensor (i.e., detection of limit and reliability). To evaluate the influence of surface area, we generated virtual models similar to SEM data, calculating and comparing surface areas within a 500 nm x 500 nm space. According to our calculations, the surface area increase due to small pores was measured at 1244% compared to a flat surface, whereas the increase due to large pores was approximately 619% (1863% - 1244% = 619%), roughly half that of small pores. This indicated that smaller pores enhance the available binding sites for nucleic acids within the same area. However, examination of fluid flow in Figure 3 revealed that nanovortex strength is higher in large pores compared to small pores. This suggests a higher chance for nucleic acids to encounter internal pore surfaces in large pores. The increased surface area implies a greater number of binding sites available for nucleic acids, while the formation of nanovortex provides increased opportunities for binding. Based on these observations, our assessment of the BSNFs-chip's effectiveness in enriching pathogens and NAs using PCR-based methods (Figure 4) was grounded in the high binding probability validated through simulations and experiments.

[Additional Resource. 3]

Moral ratio of Sal⁻/CTA⁺ = 0.85

Moral ratio of Sal⁻/CTA⁺ = 1.7

Moral ratio of Sal⁻/CTA⁺ = 3.4

200 nm

Additional Resource. 4. SEM images of various nanostructures, the first layer showing different porosity by changing the molar ratio of anionic salicylate ions (Sal-) to cationic cetyltrimethylammonium ions (CTA⁺).

3. “However, the BSNF, with its enhanced slip flow, can decrease this distance to 22.5 μm (approximately 61% reduction), indicating that the targets can approach and be captured by the BSNF more easily.”

a. Regarding the particle proximity test, please discuss the reason of using fluorescent particles with 10 μm diameter. It seems difficult to infer any effect of the surface (300 nm) on the particle flow due to the large difference of scale.

b. Please discuss compared to the nanostructures size, the decrease in distance to 22.5 μm is not much significant. It is as far as the initial condition without the features in the system. The scales of particles and features should be similar so that a meaningful decreasing in distance can be observed.

Answer. We thank the reviewer for the comment regarding the particle proximity test. The followings have explained the results in comprehensive details. First, as the reviewer pointed out, 10 μm diameter particles cannot trace fluid flows in or right above the nanoscale porous structures. However, they can draw fluid streamlines and the boundary layer near the surface qualitatively, which will be changed according to the slip length and/or permeability of the surface. Indeed, the bulk flow profile were totally shifted as the slip length is increased by just adding a thin nanoporous layer on the surface (Fig. 3 in the manuscript). Therefore, as the previous works did (ref. 24 in the manuscript), measuring how much the 10 μm particle can go closer to the surface allows us to identify the degree of slip velocity, permeability of the porous surface, and corresponding thin boundary layer which will bring particles (targets) to the surface more easily. Second, we fully agreed that the smaller trace particles may show us more clear vision of the effects of the nanoporous structures. However, tracking nanoparticles require more specialized method, because a conventional fluorescent microscope (as we used in this experiment) has a limited spatial resolution to detect them¹. Given your comment, we clarified and added these discussions and limits of the particle proximity test in the revised manuscript.

[Revised main manuscript, page 9]

“When we propel the 10 μm fluorescent particles in the solution toward a vertically aligned solid surface under perfect slip conditions with an ideally inviscid flow, the streamline (or particle trajectories) shows a two-dimensional stagnation flow with a single stagnation point at the center of the solid surface. While microscale particles cannot trace fluid flows in or right above the nanoscale porous structures, they can draw fluid streamlines and the boundary layer near the surface qualitatively, which will be changed according to the slip length and/or permeability of the surface.”

[Reference]

1. Kazoe, Y., Shibata, K. & Kitamori, T. Super-Resolution Defocusing Nanoparticle Image Velocimetry Utilizing Spherical Aberration for Nanochannel Flows. *Anal. Chem.* **93**, 13260-13267 (2021).

Reviewer #2

In this paper the authors developed a biporous silica nanofilms chip for pathogen and nucleic acid enrichment and isolation. They coupled their BSNFs-chip with the luminescence resonance energy transfer (LRET) assay for PCR-free analysis of COVID19 samples and demonstrated that their method achieved a 100-fold lower LOD. The paper can be considered for publication in Nature Communications after revisions based on comments below.

- We thank the reviewer for reviewing our manuscript and providing detailed suggestions that helped us improve the quality of the manuscript.

1) It is not clear how the chip isolate SARS-Cov-2 virus specifically? Amine-modified BSNF chip can also capture any other viruses or vesicles electrostatically. They need to clearly explain and demonstrate specificity using non target similar viruses.

Answer. We thank the reviewer for the comment regarding the specificity of our BSNFs-chip in isolating SARS-CoV-2. As the reviewer correctly noted, the amine-modified BSNFs-chip is designed to electrostatically capture a range of entities with a negative surface charge, including cells, bacteria, viruses, and vesicles. This capability is exemplified in Fig. 4 of our manuscript, where the HCT116 cell line and SARS-CoV-2 virus are used. The broad-spectrum capture capability of the BSNFs-chip is a significant feature, enabling applications for the enrichment of target biomaterials in biological sample processing. **Supplementary Fig. 10** has been revised to clearly explain this broad-spectrum capture capability.

However, the specific detection of SARS-CoV-2 in our system is achieved through the integration of the PCR-free LRET assay, which plays a role in conferring specificity to our detection system. This assay is designed to target the S gene of SARS-CoV-2, thereby ensuring accurate and specific detection. To address the reviewer's concern about specificity, we conducted additional experiments with viruses similar to SARS-CoV-2, but not targeted by our system (**Supplementary Fig. 16a,b**). These results were instrumental in showing that the LRET assay can distinctly identify SARS-CoV-2 RNA, effectively differentiating it from other viral RNAs. This specificity is crucial for the accurate detection of SARS-CoV-2, ensuring that our system is not confounded by the presence of other viruses.

Furthermore, we recognize the potential of the BSNFs-chip in enriching and isolating vesicles which also carry a negative surface charge. We plan to explore this application in future research, expanding the utility of the BSNFs-chip beyond its current focus on SARS-CoV-2 detection. This detailed information, including the experimental details and results demonstrating the specificity of the LRET assay, has been added to the revised manuscript.

[Revised main manuscript, page 14]

"Future investigations will focus on extending the applicability of these nanomaterials through exploring the micelle aggregation mechanism and precise pore control, including the design of customized nanostructures, while simultaneously aiming to enhance the integration of the BSNFs-chip with the LRET assay to expand their combined utility beyond SARS-CoV-2 detection to a broader range of pathogens. We anticipate that this approach will play an impactful role in advancing the field of rapid and sensitive diagnostic methods for infectious diseases in the years to come."

[Revised main manuscript, page 12]

"Furthermore, the DNA oligos on the surface of the LRET donor and acceptor could hybridize specifically to SARS-CoV-2 RNA with no obvious cross-reactivity to other non-target NAs (Supplementary Fig. 14a). We validated the specificity of the LRET assay using 1 pM of non-target sequences of other common contagious respiratory viruses. The LRET assay successfully distinguished SARS-CoV-2 from human coronavirus OC43 (hCoV-OC43), hCoV-NL63, hCoV-229E, and influenza A virus (IAV) (Supplementary Fig. 14b)."

[Revised main manuscript, page 22]

“The specificity of the LRET assay was determined using target RNA (SARS-CoV-2) and non-target RNAs including human coronavirus OC43 (hCoV-OC43), human coronavirus 229E (hCoV-229E), human coronavirus NL63 (hCoV-NL63), and H3N2 Influenza A virus (IAV). The hCoV-OC43 and hCoV-229E RNAs were provided by the Korea Bank for Pathogenic Viruses (Seoul, Korea). The hCoV-NL63 RNA was provided by the National Culture Collection for Pathogens (Cheongju, Korea). H3N2 IAV (A/Brisbane/10/2007) RNA was provided by the Korea Research Institute of Bioscience and Biotechnology (KRIBB, Daejeon, Korea).”

[Revised Supplementary Fig. 10]

Supplementary Fig. 10. Schematic overviews of the sample preparation process of chips with **a**, BSNFs-chip and **b**, Flat-chip. The amine-modified BSNFs and flat surfaces are designed to electrostatically capture a range of entities with a negative surface charge, including cells, bacteria, viruses, and vesicles.

[Revised Supplementary Fig. 16]

Supplementary Fig. 16. LRET-based detection of SARS-CoV-2 RNA. **a**, Schematic illustration of the LRET-based detection specific to SARS-CoV-2 RNA. **b**, The LRET assay demonstrated specificity for SARS-CoV-2, with no cross-reactivity towards other common respiratory viruses, including human coronavirus OC43 (hCoV-OC43), hCoV-NL63, hCoV-229E, and influenza A virus (IAV). **c**, Emission spectra of the LRET donor with various concentrations of SARS-CoV-2 RNA. **d**, Calibration curve of quenching efficiencies with increasing concentrations of SARS-CoV-2 RNA. A linear fit to the data over this concentration range yielded the relation $(I_0 - I_x)/I_0 = 0.065 \times \log C + 0.094$, where C is the concentration of RNA, with a correlation coefficient $R^2 = 0.998$. All reported values represent the mean \pm SD, $n = 3$. The cut-off value was determined by applying optimal combinations of clinical sensitivity and specificity from ROC curve based on the Youden index point. **e**, Luminescence lifetime curves at 800 nm of LRET donor in the absence or presence of target RNA (under 980 nm excitation).

2) There is a similar concern about for nucleic acids. Any nucleic acids can also be trapped electrostatically at the amin-modified chips. Specificity of the chip is in question.

Answer. We thank the reviewer for the comment regarding the specificity of our BSNFs-chip in trapping nucleic acids (NAs) from SARS-CoV-2. Building on our previous response, the amine-modified BSNFs-chip demonstrates a robust capability for capturing various NAs through electrostatic and covalent binding. This capability is evidenced in our study, particularly in Fig. 4, where we successfully captured both genomic DNA and RNA from the HCT116 cell line, as well as viral RNA from SARS-CoV-2. Notably, the specificity of our system in detecting SARS-CoV-2 hinges on the PCR-free LRET assay. This assay is designed to target the S gene of SARS-CoV-2, thereby ensuring accurate and specific detection. In the revised manuscript, we have included experimental data and results that underscore the LRET assay's effectiveness in identifying SARS-CoV-2 RNA, distinguishing it from other viral RNAs. These details, which are elaborated upon in our response to the first comment, have been explained into the revised manuscript to provide a comprehensive understanding of our system's specificity in detecting SARS-CoV-2.

3) What is the total time required from sample collection to get the answer?

Answer. We thank the reviewer for the comment regarding the total time required for our process, from sample collection to get the answer. Our study demonstrates that the use of the BSNFs-chip for pathogen and NA enrichment/isolation takes approximately 40 minutes. For the detection

of SARS-CoV-2 in clinical samples, our methodology incorporates a sequential procedure. First, we employ the BSNFs-chip for pathogen and NA enrichment/isolation, which takes around 40 minutes using a 200 μ L sample volume. Following this, the PCR-free LRET assay is conducted, requiring an additional 10 minutes. Therefore, the total time from sample collection to result acquisition for SARS-CoV-2 detection is approximately 50 minutes. This streamlined process, combining the BSNFs-chip with the PCR-free LRET assay, not only enhances the efficiency of pathogen detection but also significantly reduces the overall time required for analysis compared to traditional methods. This comprehensive timing information has been detailed in the revised manuscript.

[Revised main manuscript, page 11]

"This strategy begins with the enrichment/isolation of pathogens and NAs using the BSNFs-chip (approximately 40 min), which is followed by the identification of enriched target RNAs using the LRET assay (approximately 10 min), and the entire process from sample collection to obtaining results for SARS-CoV-2 detection is completed within 50 minutes (Fig. 5a)."

4) It is good that their chip can be reused. Is there any change in performance after reuse? How many times a chip can be reused?

Answer. We thank the reviewer for the comment regarding the reusability of our chip. The reusability of the BSNFs-chip represents a significant advancement in terms of both cost-effectiveness and environmental sustainability. In the manuscript, Supplementary Fig. 13 provides a comprehensive analysis of the BSNFs-chip's reusability. Figure S13a presents SEM images demonstrating the structural stability of the BSNF after the sample preparation process, showing that the BSNF maintains its integrity even after sample processing. Figure S13b further analyzes the structural robustness and reusability of the chip in pathogen and NA enrichment/isolation. The Ct values obtained were 27.78 ± 0.13 for the first use, 27.3 ± 0.35 for the second use, and 27.03 ± 0.30 for the third use, indicating high reproducibility. As depicted in these figures, our findings suggest that the BSNFs-chip could potentially be used more than three times, given its robustness and consistent performance. However, we chose to use the BSNFs chip only once to completely avoid contamination risks. This decision was made despite the potential for extended reuse, as our priority was to ensure the highest standards of accuracy in every application. While the chip exhibits excellent structural stability and efficiency, we prioritized contamination control over the possibility of reuse. This approach and the associated data have been added to the revised manuscript.

[Revised main manuscript, page 11]

"The structural stability of the BSNFs-chip was also confirmed even after being used for pathogen and NA enrichment/isolation (Supplementary Fig. 13). This robustness makes the BSNFs-chip suitable for both disposable and reusable applications. Notably, the chip demonstrated consistent performance across multiple uses, with Ct values of 27.78 ± 0.13 for the first use, 27.3 ± 0.35 for the second use, and 27.03 ± 0.30 for the third use, indicating high reproducibility. These results further establish the BSNFs-chip as a promising sample preparation platform for the accurate diagnosis of infectious diseases, highlighting its potential as a reliable tool in the sensitive and precise diagnosis of infectious diseases."

[Revised main manuscript, page 20]

"All used microfluidic chips were disposed of after a single use to avoid contamination risks, and the eluted NAs were stored at either -20 or -80 °C for future use."

5) How long the chip is stable? How stable the integrated chip when coupled with LRET assay?

Answer. We thank the reviewer for the comment regarding the stability of BSNFs-chip. In response to the reviewer's comment, we fabricated a new BSNFs-chip and assessed its stability over a two-week period. The stability of BSNF chips can offer a significant advantage in terms of convenience in storage and ensuring reliability and consistency of experimental results. The BSNFs-chips have been utilized under prolonged storage at room temperature post-fabrication. These conditions are consistent across experiments, notably in sample preparation for PCR test and LRET assay. We conducted SEM imaging and FT-IR spectroscopy over a two-week period immediately following chip fabrication, confirming structural integrity and the consistent presence of amine peak around 3350 cm^{-1} . The detailed information, including experimental results demonstrating the stability of BSNFs-chip, has been added in **Supplementary Fig. 12**, as described in the revised manuscript.

[Revised main manuscript, page 11]

“The stability of the BSNFs-chip was confirmed over a two-week period immediately following chip fabrication through SEM imaging and FT-IR spectroscopy (Supplementary Fig. 12).”

[Revised Supplementary Fig. 12]

Supplementary Fig. 12. SEM images and FT-IR spectrums showing the stability of the BSNFs-chip over a two-week period immediately following chip fabrication.

The BSNFs-chip was utilized underwent prolonged storage at room temperature post-fabrication. These conditions were consistent across experiments, notably in sample preparations for PCR test and LRET assay.

6) Their chip could analyze 10-fold diluted samples. What is performance of the chip if they work with undiluted sample. Any device that can analyze undiluted samples are preferable.

Answer. We thank the reviewer for the comment regarding the performance of our BSNFs-chip with undiluted samples. We apologize for any confusion caused by a wording error in the Methods section, specifically, “10-fold dilutions of SARS-CoV-2 RNA solution (15 µl) were mixed with ...”. We performed analyses using clinical samples in their native, undiluted state, as detailed in Fig. 5 and Supplementary Figs. 17 and 18. Notably, Supplementary Fig. 17 illustrates the qRT-PCR detection of SARS-CoV-2 RNA from nasopharyngeal swab samples, confirming the ability of the chip to process and detect the virus directly from clinical specimens. Additionally, 15 µl of these samples were mixed with LRET donor, acceptor, and reaction buffer, constituting 10% of the total reaction solution. However, it is noted that the samples were not further diluted to assess the performance of the BSNF chip integrated with the LRET assay. The comparisons in Supplementary Figs. 18a and 18b between our BSNFs-chip and traditional extraction methods, particularly using the LRET assay, further underscore the chip's enhanced sensitivity and reliability with undiluted samples. These findings collectively establish the BSNFs-chip's suitability for practical diagnostic use in clinical settings, confirming its adaptability and effectiveness in handling various sample conditions, including undiluted samples. To help clear understanding, the phrase “10-fold dilutions of” was deleted and a detailed method for LRET-based detection using clinical samples was described in the revised manuscript.

[Revised main manuscript, page 21]

~~10-fold dilutions of~~ SARS-CoV-2 RNA solution (15 µl), which was isolated by the BSNFs-chip or QIAamp Viral RNA Mini Kit (Cat no. 52906, Qiagen) were mixed with the LRET donor (2 µg) and the LRET acceptor (10 pmol, DNA modified IR800 dye) (Integrated DNA Technology, IDT) in HEPES buffer (10 mM, pH 6.2) and incubated at room temperature with 600 rpm shaking for 10 minutes.

[Revised main manuscript, page 21]

We validated the clinical applicability of the LRET assay using 30 clinical samples including 20 COVID-19 positive patients and 10 healthy controls. 15 µl of viral RNA samples isolated by the QIAamp Viral RNA Mini Kit (Qiagen) and the BSNFs-chip were mixed with the LRET donor (2 µg) and the LRET acceptor (10 pmol) in HEPES buffer (10 mM, pH 6.2) (total volume = 150 µl). Then, the LRET-based SARS-CoV-2 RNA detection was performed as described above.

Reviewer #3

This manuscript was entitled with "Biporous silica nanostructure-induced nanovortex in a microfluidic channel for nucleic acid enrichment, isolation, and PCR-free detection". It introduces a biporous silica nanofilms-embedded sample preparation chip (BSNFs-chip) for pathogen and NA enrichment/isolation. This design of BSNFs-chip improves the performance through enhancing the surface area and promote the formation of nanovortex, and showing good performance in the clinical results. Therefore, I recommend this work publication after undergoing a major revision focusing on issues like:

- We thank the reviewer for reviewing our manuscript and providing detailed suggestions that helped us improve the quality of the manuscript.

1. This design of nanostructures plays a critical role in NA analysis, comprising large and small pore layers. Why design two kinds of pore structures?

Answer. We thank the reviewer for the comments regarding the control of porosity (area) in fabrication protocol and the sensor performance. Accordingly, we explored the nanostructure porosity by changing the ratio of the surfactants, as demonstrated in **Additional Resource. 1**. The following sections have explained the results in comprehensive details.

The porosity of the nanostructures can be controlled by altering the micelle template sizes which is controlled by changing the ratio of the surfactants with opposite charges. In the case of small pores, the micelle template possesses a more positive surface charge, facilitating better interaction with the O₂ plasma-treated negatively charged surface, which results in a well-formed initial layer. However, synthesis of films with larger pore sizes greater than 60 nm presented challenges, as the surface charge of the micelle template neutralized due to increased intrusion of anionic salicylate ions into the cationic CTAC micelles, subsequently weakening its interaction with the substrate surface. However, we found that leveraging a first layer formed by small pores acted as a seed layer during sol-gel process, facilitating the formation of large pores in the subsequent layer—an approach we employed in our design. Therefore, to compare the effect of porosity (surface) on the sensor performance, we synthesized the film with small pores and the film with both small pores and large pores.

In our study, the increased surface area and the generation of nanovortex resulting from the chip's internal porous nanostructures significantly impacted the preparation and enrichment of nucleic acids. This purified and concentrated nucleic acids improve the performance of the sensor (i.e., detection of limit and reliability). To evaluate the influence of surface area, we generated virtual models similar to SEM data, calculating and comparing surface areas within a 500 nm x 500 nm space. According to our calculations, the surface area increase due to small pores was measured at 1244% compared to a flat surface, whereas the increase due to large pores was approximately 619% (1863% - 1244% = 619%), roughly half that of small pores. This indicated that smaller pores enhance the available binding sites for nucleic acids within the same area. However, examination of fluid flow in Figure 3 revealed that nanovortex strength is higher in large pores compared to small pores. This suggests a higher chance for nucleic acids to encounter internal pore surfaces in large pores. The increased surface area implies a greater number of binding sites available for nucleic acids, while the formation of nanovortex provides increased opportunities for binding. Based on these observations, our assessment of the BSNFs-chip's effectiveness in enriching pathogens and NAs using PCR-based methods (Figure 4) was grounded in the high binding probability validated through simulations and experiments.

[Additional Resource. 1]

Additional Resource. 1. SEM images of various nanostructures, the first layer showing different porosity by changing the molar ratio of anionic salicylate ions (Sal⁻) to cationic cetyltrimethylammonium ions (CTA⁺).

2. In this work, the two BSNFs had a total height of approximately 800 nm within the microchannel, and occupied only 0.27% of the total microchannel height of 300 μm . Why chose this height? and this design gives most of sample no chance to contact with 3aminopropyl(diethoxy) methylsilane. Please compare the effects of different BSNFs heights on sample preparation.

Answer. We thank the reviewer for the comment regarding the influence of dimension selection on sensor performance. In response to the reviewer's comment, we investigated the growth of nanostructures and compared simulations using different heights, as detailed in the following sections.

We observed the growth of nanostructures over reaction time, layer by layer, from small pores in the first layer to large pores in the second layer. These nanostructures displayed gradual growth over the reaction time. However, the small pores in the first layer ceased growth after 3 hours, reaching a maximum height of about 145 nm. Subsequently, they began filling from the bottom, resulting in a reduction of the effective pore channel height. Similarly, the large pores in the second layer maintained their maximum height of 400 nm after 2 hours without further growth. Consequently, we verified that the maximum achievable height for the biporous silica nanofilm was 400 nm (800 nm within the microchannels). The detailed information, including experimental results demonstrating the growth of BSNF, has been added in **Supplementary Fig. 6**, as described in the revised manuscript.

An increase in film height has been anticipated to potentially enhance surface area, offering increased opportunities for molecular binding. However, additional simulations (**Additional Resource. 2**) revealed that increased height did not proportionally correlate with an improvement in chip performance. Specifically, applying a flow rate of 100 $\mu\text{l}/\text{min}$ to simple square-shaped pore structures with heights of 500, 1000, and 1500 nm, demonstrated a limitation in dynamic flow depth as the height increased, indicating inefficiency. Therefore, based on this study, we conclusively determined the optimal nanostructure height at 400 nm, capped at 800 nm within the microchannel, considering its direct implications on sensor performance in fluid analysis. **Additional resource. 2** related to this result has been added below.

[Revised main manuscript, page 7] "The cross-view SEM image of the BSNF reveals a total height of approximately 400 nm, with a distinct boundary between the first (height approximately 145 nm) and second layers (height approximately 255 nm) (Fig. 2c and Supplementary Fig. 6)."

[Revised Supplementary Fig. 6]

Supplementary Fig. 6. Cross-sectional SEM images of **a**, PSNF and **b**, BSNF over reaction time.

These nanostructures displayed gradual growth over the reaction time. However, the small pores in the first layer ceased growth after 3 hours, reaching a maximum height of about 145 nm. Subsequently, they began filling from the bottom, resulting in a reduction of the effective pore channel height. Similarly, the large pores in the second layer maintained their maximum height of 400 nm after 2 hours without further growth.

[Additional Resource. 2]

Additional Resource. 2. 3D images of flow velocity field in a simple square-shaped pore structures with heights of **a**, 500 nm, **b**, 1000 nm, and **c**, 1500 nm at a flow rate of 100 $\mu\text{l}/\text{min}$.

3. The numerical simulation of fluid should no reduce scale (1/100), because it masks the real situation, amplifies the effect of the BSNFs.

Answer. We appreciate the reviewer's comment regarding the scale reduction in numerical simulations. As mentioned in the Supplementary Fig.1 and Supplementary Table 1, we compared the effects of the BSNF at a reduced scale of 1/100. Despite intending to conduct simulations at the actual scale as suggested by the reviewer, hardware constraints on our computer posed challenges (**Additional Resource. 3**). As an alternative, we carried out calculations at a reduced 1/10 scale, which proved to be computationally feasible. Interestingly, both the 1/10 and 1/100 reduced scales showed similar interstitial velocity values (**Additional Resource. 4**). This indicates that the 1/100 scale doesn't depict the impact of BSNF differently. However, explaining simulations at the 1/10 reduced scale presented visualization challenges within our computer specifications. Therefore, considering these outcomes, we opted for the 1/100 reduced scale to represent visually the effects of BSNF without exaggeration. Additional resources related to this result have been added below.

[Additional Resource. 3]

```
-----  
System Information  
-----  
Time of this report: 11/24/2023, 18:50:46  
Machine name: DESKTOP-LGUVQ7G  
Machine Id: {36A8B246-6DD6-4D53-9656-93B4863990BB}  
Operating System: Windows 11 Pro for Workstations 64-bit (10.0, Build 22621) (22621.ni_release.220506-1250)  
Language: Korean (Regional Setting: Korean)  
System Manufacturer: HP  
System Model: HP Z6 G4 Workstation  
BIOS: P60 v02.91 (type: UEFI)  
Processor: Intel(R) Xeon(R) Silver 4214R CPU @ 2.40GHz (48 CPUs), ~2.4GHz  
Memory: 262144MB RAM  
Available OS Memory: 261848MB RAM  
Page File: 20780MB used, 361993MB available  
Windows Dir: C:\windows  
DirectX Version: DirectX 12  
DX Setup Parameters: Not found  
User DPI Setting: 96 DPI (100 percent)  
System DPI Setting: 96 DPI (100 percent)  
DWM DPI Scaling: Disabled  
Miracast: Available, no HDCP  
Microsoft Graphics Hybrid: Not Supported  
DirectX Database Version: 1.4.7  
DxDiag Version: 10.00.22621.0001 64bit Unicode  
  
-----  
Display Devices  
-----  
Card name: NVIDIA RTX A4000  
Manufacturer: NVIDIA  
Chip type: NVIDIA RTX A4000
```

Additional Resource. 3. Capture image showing system information and display devices used in numerical simulations.

[Additional Resource. 4]

Additional Resource. 4. A graph comparing layered interstitial velocity at scales reduced to 1/10 and 1/100.

4. The sample preparation chips effectively concentrated each cell and virus sample from 1 ml to a final volume of 100 μ l. However, the detection limitation had 10-fold lower LOD. It means that this chip only played a role in concentration, thus it should compare different concentration (for example magnetic concentration in the microfluidic) method. and this result demonstrated that the pore structures had a little influence on detection performance.

Answer. We thank the reviewer for the comment regarding the 10-fold lower LOD of our BSNFs-chip. As illustrated in Fig. 4c of our manuscript, for DNA, both the BSNFs- and PSNFs-chips showed a 10-fold lower LOD (1×10^1 cells/ml) compared to the Flat-chip, and a 100-fold lower LOD

than conventional methods. For RNA, encompassing both genomic RNA and viral RNA, the BSNFs-chip exhibited a 10-fold higher sensitivity (LOD of 1×10^1 cells/ml for genomic RNA and 0.96×10^0 PFU/ml for viral RNA) than the Flat- and PSNFs-chips, and a 100-fold higher sensitivity compared to conventional methods, as detailed in Fig. 4d and 4e. This result is not simply due to using a larger sample volume to achieve a 10-fold improvement in sensitivity. Instead, it underscores the BSNFs-chip's capability to surpass the limitations of detection methods at low concentration levels. This is evidenced by its ability to achieve sensitivity levels 10 to 100 times greater than those of conventional methods, a significant advancement attributable to its unique biporous silica nanostructures and the nanovortex effect. These innovative design features are crucial for detecting pathogens and NAs at substantially lower concentrations than possible with conventional methods.

5. For LRET-based viral RNA detection, the linear detection range should be provided.

Answer. Thanks for the Reviewer's attentive comments. The LRET assay demonstrated a high correlation coefficient (R^2) of 0.99783 ranging from 10^{-1} to 10^3 PFU of SARS-CoV-2 RNA, as shown in Supplementary Fig. 16d. Following the Reviewer's comment, we have added more detail on the linear detection range in the revised manuscript.

Supplementary Fig. 16 c, Emission spectra of the LRET donor with various concentrations of SARS-CoV-2 RNA. **d**, Calibration curve of quenching efficiencies with increasing concentrations of SARS-CoV-2 RNA. A linear fit to the data over this concentration range yielded the relation $(I_0 - I_x)/I_0 = 0.065 \times \log C + 0.094$, where C is the concentration of RNA, with a correlation coefficient $R^2 = 0.998$.

[Revised main manuscript, page 12]

The relative intensities displayed a linear relationship with the logarithmic concentration of SARS-CoV-2 RNA ranging from 10^{-1} to 10^3 PFU, with a high correlation coefficient (R^2) of 0.998 (Fig. 5c and Supplementary Fig. 16c,d).